# Backpropagation-Free Test-Time Adaptation via Probabilistic Gaussian Alignment

Youjia Zhang[1]    Youngeun Kim[2] *    Young-Geun Choi[1]
Hongyeob Kim[1]    Huiling Liu[1]    Sungeun Hong[1][†]

[1]Sungkyunkwan University    [2]Amazon
https://aim-skku.github.io/ADAPT

## Abstract

Test-time adaptation (TTA) enhances the zero-shot robustness under distribution shifts by leveraging unlabeled test data during inference. Despite notable advances, several challenges still limit its broader applicability. First, most methods rely on backpropagation or iterative optimization, which limits scalability and hinders real-time deployment. Second, they lack explicit modeling of class-conditional feature distributions. This modeling is crucial for producing reliable decision boundaries and calibrated predictions, but it remains underexplored due to the lack of both source data and supervision at test time. In this paper, we propose **ADAPT**, an **A**dvanced **D**istribution-**A**ware and back**P**ropagation-free **T**est-time adaptation method. We reframe TTA as a Gaussian probabilistic inference task by modeling class-conditional likelihoods using gradually updated class means and a shared covariance matrix. This enables closed-form, training-free inference. To correct potential likelihood bias, we introduce lightweight regularization guided by CLIP priors and a historical knowledge bank. ADAPT requires no source data, no gradient updates, and no full access to target data, supporting both online and transductive settings. Extensive experiments across diverse benchmarks demonstrate that our method achieves state-of-the-art performance under a wide range of distribution shifts with superior scalability and robustness.

## 1   Introduction

While Vision-language models (VLMs) such as CLIP [39] can recognize diverse visual concepts using only class names, their robustness drops significantly when test-time distributions differ from those seen during pre-training [33, 55]. Test-time adaptation (TTA) offers a promising solution to enhance the robustness of VLMs by adapting them using only unlabeled test data [49]. From a distributional perspective, TTA helps correct the mismatch between the class-conditional feature distributions learned during pretraining and those encountered at test time. Early approaches, including prompt tuning [33, 62, 31, 42] and adapter tuning methods [11, 68, 56, 46, 61], refine additional prompts or adapter layers through iterative optimization. While more parameter-efficient than full fine-tuning, these methods still depend on costly backpropagation, hindering their deployment in real-time or streaming scenarios [63].

Recent advances in TTA have increasingly emphasized efficiency, leading to the development of training-free approaches. Several methods leverage memory banks constructed from test samples [21, 64] or few-shot training data to adapt classifiers based on sample similarities [63, 65], but often

---

*This work was completed prior to the author joining Amazon

[†]Corresponding author: Sungeun Hong (csehong@skku.edu)

39th Conference on Neural Information Processing Systems (NeurIPS 2025).

Table 1: Comparison with existing TTA methods.

| Method | BP-Free | Distribution-Aware | Task Setting | |
|---|---|---|---|---|
| | | | Online | Transductive |
| Prompt Tuning [33, 62, 31, 42, 54] | ✗ | ✗ | ✓ | ✗ |
| Adapter Tuning [11, 68, 56, 46, 61] | ✗ | ✗ | ✓ | ✗ |
| Similarity Score [21, 64, 65, 63, 47] | ✓ | ✗ | ✓ | ✗ |
| Transductive Learning [20, 29, 59, 69, 51] | ✓ | ✓ | ✗ | ✓ |
| ADAPT (Ours) | ✓ | ✓ | ✓ | ✓ |

[*]*BP-Free*: No backpropagation at test time.

require tuning task-specific fusion coefficients, limiting their generalizability across tasks. Other methods [20, 29] propagate labels from text prompts to test samples via static graph construction, yet frequently depend on external datasets and involve costly iterative updates.

Alongside these trends, transductive learning [66] has also gained attention. This setting assumes full or partial access to the target data during inference, enabling the model to capture the global structure of the test distribution. It works well in offline scenarios such as batch processing of medical scans or document classification, where the model can benefit from support sets or feature clusters. However, such methods often require complete access to target data [59, 69, 60, 58], or even source data [51, 20], making them unsuitable for strict online or streaming environments where samples arrive and are processed sequentially, one by one, without access to future data.

Despite advances in efficiency and data accessibility, existing TTA approaches still overlook how class-conditional feature distributions are structured. Many rely solely on text-based prototypes to define decision boundaries [33, 46, 61, 11, 42] or adjust similarity scores using task-specific scaling [21, 64, 65, 63, 47], without explicit class distribution modeling. Moreover, since TTA operates without supervision on the target domain and lacks access to source data, estimating class distributions remains challenging. As a result, existing methods often fail to capture intra-class variation and inter-class confusion (see Figure 3), which leads to unstable decision boundaries and reduced robustness under distribution shifts.

In this paper, we introduce **ADAPT**, an **A**dvanced **D**istribution-**A**ware method for back**P**ropagation-free **T**est-time adaptation. Our approach reformulates test-time adaptation as a probabilistic inference task by explicitly modeling class-conditional feature distributions under a Gaussian assumption. Through the use of high-confidence historical predictions, we progressively estimate class means and a shared covariance matrix without relying on supervision or source data. This enables a closed-form, training-free solution that supports both online and transductive settings, without requiring additional training or task-specific tuning (see Table 1). Notably, our closed-form approach enables test-time adaptation via explicit, non-iterative formulas derived from probabilistic assumptions, without any iterative optimization or backpropagation. ADAPT provides surprisingly strong, backpropagation-free decision boundaries and achieves efficient and robust test-time adaptation.

Our contributions are summarized as follows:

- We propose ADAPT, a backpropagation-free test-time adaptation framework that models class-conditional feature distributions under a Gaussian assumption, enabling one-pass, closed-form adaptation without iterative optimization.

- We initialize the class means from CLIP prototypes and update them using high-confidence features stored in fixed-size knowledge banks, enabling source-free, training-free adaptation in both online and transductive settings.

- Extensive experiments across diverse tasks and multiple benchmarks show that ADAPT outperforms prior state-of-the-art methods under distribution shifts. Our method also offers strong scalability, faster inference, and stable performance across various test-time conditions.

## 2 Related Works

**Online Test-Time Adaptation of VLMs.** Online TTA adapts pre-trained VLMs to downstream tasks in streaming scenarios. Existing approaches fall into two main directions: prompt tuning and

adapter tuning. Prompt-based methods [33, 31, 42, 71, 62, 9, 55] adapt prompts by minimizing entropy across augmented views [33, 9, 31] or aligning multi-modal features using proxy data [42]. Adapter-based approaches [11, 61, 46] inject lightweight modules into VLMs while keeping the backbone frozen. Despite their effectiveness, these methods rely on backpropagation, incurring high computation and memory costs. Recently, training-free TTA methods have gained attention for their efficiency. Approaches like Tip-Adapter [63] and SuS-X [47] utilize class-wise support samples from external sources for nearest-neighbor matching. Others [21, 65, 64] adapt predictions by retrieving historical features based on sample similarity. While gradient-free, they often require careful hyperparameter tuning and lack explicit modeling of class distributions, making them sensitive to distribution shifts and prone to unstable decision boundaries. We propose a distribution-aware and propagation-free TTA paradigm, achieving stable and efficient zero-shot performance in challenging scenarios involving severe distribution shifts and fine-grained tasks.

**Transductive Zero-Shot Learning in VLMs.** Transductive learning [19, 66], where all test samples are accessible during inference, was initially explored in the few-shot learning literature for leveraging both labeled and unlabeled samples to improve generalization. By modeling the structure of the test set, it often outperforms inductive methods [30, 72, 2]. Inspired by these successes, recent efforts have extended transductive learning to the multimodal domain to improve the zero-shot adaptation of VLMs. ZLaP [20] and ECALP [29] build similarity graphs over test-time features to propagate labels but rely on costly iterative refinement steps. Wang *et al.* [51] employs a Gaussian Discriminant Analysis classifier by estimating distributional statistics from external source data. Frolic [69] adjusts predictions by learning class-wise prompt distributions inferred from target data moments. TransCLIP [59], Histo-TransCLIP [60] and StatA [58] model each class with a multivariate Gaussian and penalize the deviations from pseudo-labels and text-driven priors. While effective, these methods often require access to all or large batches of test data, limiting their use in online or streaming settings. To overcome this, we propose an adaptive method that builds class-wise Gaussian distributions using reliably accumulated high-confidence predictions. This supports both transductive and online adaptation, offering strong deployment flexibility.

# 3 Method

## 3.1 Preliminaries

**Zero-Shot Classification with VLMs.** Vision-language models, such as CLIP, consist of two parallel encoders: an image encoder $\theta_v(\cdot)$ and a text encoder $\theta_t(\cdot)$. Given a set of $K$ candidate classes $\{t_k\}_{k=1}^K$ and unlabeled images $\{x_i\}_{i=1}^N$, their embeddings are computed as $\mathbf{x}_i = \theta_v(x_i) \in \mathbb{R}^d$ and $\mathbf{t}_k = \theta_t(t_k) \in \mathbb{R}^d$. The text-derived class prototypes $\mathbf{t}_k$ serve as fixed semantic anchors that define implicit decision boundaries in the shared embedding space. Zero-shot predictions are made by computing the cosine similarity between image $\mathbf{x}_i$ and class prototypes $\mathbf{t}_k$, scaled by parameter $\tau$:

$$\hat{y}_{i,k} = \frac{\exp(\mathbf{t}_k^\top \mathbf{x}_i / \tau)}{\sum_{j=1}^K \exp(\mathbf{t}_j^\top \mathbf{x}_i / \tau)}. \tag{1}$$

**Constructed Knowledge Banks.** To enable effective test-time adaptation without access to source data, we maintain a set of class-wise knowledge banks $\mathcal{B} = \{\mathcal{B}_k\}_{k=1}^K$, where each $\mathcal{B}_k$ stores a small set of high-confidence samples associated with pseudo-class $k$. These banks act as lightweight memory modules, continuously accumulating reliable evidence from previously seen test samples. To determine which samples are sufficiently reliable for inclusion, we compute a confidence score for each test input $\mathbf{x}_i$ based on the negative entropy of its CLIP-derived prediction $\hat{\mathbf{y}}_i$:

$$\text{Conf}(\mathbf{x}_i) = \sum_{k=1}^K \hat{y}_{i,k} \log \hat{y}_{i,k}. \tag{2}$$

We maintain a fixed-size buffer $\mathcal{B}_k$ with capacity $L$ ($L \ll N$) for each class. The impact of $L$ is discussed in Section 4.3. A new test sample $\mathbf{x}_i$ with pseudo-class $k$ is inserted only if $\mathcal{B}_k$ is not full or its confidence score exceeds the lowest-confidence entry in $\mathcal{B}_k$. This threshold-free, priority-based mechanism ensures that the banks remain fresh, compact, and focused on the most informative observations. By adaptively collecting representative and high-confidence samples without supervision or backpropagation, the knowledge banks provide a reliable basis for downstream estimation of test-time class-conditional distributions.

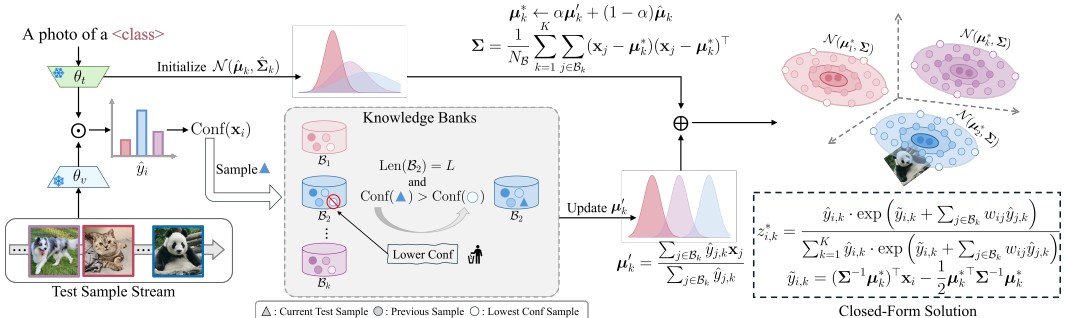

Figure 1: Overview of Online ADAPT. We perform TTA by modeling class-conditional feature distributions under a Gaussian assumption with shared covariance across classes. Class means are initialized from CLIP prototypes and refined using high-confidence samples in fixed-size per-class knowledge banks. To avoid error accumulation, the current test sample is excluded from updates. Predictions are made via a closed-form, backpropagation-free solution. In the transductive setting, the knowledge bank is built using the top-$L$ most confident samples per class from the full test set.

## 3.2 ADAPT: Backpropagation-free and Distribution-aware Test-time Adaptation

**Backpropagation-free TTA via Gaussian Discriminant Analysis.** Gaussian Discriminant Analysis (GDA) is a classical generative model that assigns class labels based on the likelihood of class-conditional distributions [13]. Recent studies have shown that CLIP features exhibit class-conditional clustering and can be well approximated by Gaussian distributions [51, 69, 59]. Motivated by these findings, we adopt GDA as a backpropagation-free probabilistic classifier for test-time adaptation (TTA). Specifically, to achieve efficient, real-time adaptation, we simply assume that features conditioned on class $k$ follow a Gaussian distribution with a shared covariance matrix:

$$\mathbb{P}_{i,k} = \mathbb{P}(\mathbf{x}_i|y_k) = \mathcal{N}(\mathbf{x}_i; \boldsymbol{\mu}_k, \boldsymbol{\Sigma}) = \frac{1}{\sqrt{(2\pi)^d|\boldsymbol{\Sigma}|}} \exp\left(-\frac{1}{2}(\mathbf{x}_i - \boldsymbol{\mu}_k)^\top \boldsymbol{\Sigma}^{-1}(\mathbf{x}_i - \boldsymbol{\mu}_k)\right), \quad (3)$$

where $\boldsymbol{\mu}_k$ and $\boldsymbol{\Sigma}$ denote the class mean and shared covariance, respectively. While this assumption may oversimplify real-world distributions, it enables closed-form inference with minimal computational overhead, making it well-suited for real-time, resource-constrained deployment.

From the Bayes' rule, the posterior probability of the $k$-th class given image $\mathbf{x}$ is $\mathbb{P}(y_k|\mathbf{x}) = \frac{\mathbb{P}(\mathbf{x}|y_k)\,\mathbb{P}(y_k)}{\mathbb{P}(\mathbf{x})} \propto \pi_k \mathcal{N}(\mathbf{x}; \boldsymbol{\mu}_k, \boldsymbol{\Sigma})$. Assuming a uniform class prior $\pi_k = \frac{1}{K}$, then the Bayes optimal prediction label for $\mathbf{x}_i$ is known as the argmax of $\tilde{y}_{i,k}$ over $k = 1, \ldots, K$:

$$\tilde{y}_{i,k} = \mathrm{w}_k^\top \mathbf{x}_i + b_k, \quad \text{where } \mathrm{w}_k = \boldsymbol{\Sigma}^{-1}\boldsymbol{\mu}_k, b_k = -\frac{1}{2}\boldsymbol{\mu}_k^\top \boldsymbol{\Sigma}^{-1}\boldsymbol{\mu}_k. \quad (4)$$

The estimation of $\boldsymbol{\mu}_k$ ($k = 1, \ldots, K$) and $\boldsymbol{\Sigma}$ can be optimization-free, since their maximum likelihood estimators under the i.i.d. assumption are just the class-wise empirical averages and the pooled sample covariance matrix of $\mathbf{x}$ (proof in Section A.1).

**Correcting Online Likelihood Bias via Constructed Knowledge Banks.** While GDA enables label prediction based on estimated class statistics, directly applying it in an online test-time setting results in biased likelihood estimates due to limited samples and overconfident early predictions. Inspired by prior works on transductive adaptation and regularized inference [72, 59, 58], we propose a bias correction framework that minimizes a regularized objective combining three components: (i) Online Negative Log-Likelihood $-z_i^\top \log \mathbb{P}_i$, (ii) CLIP Prior-based Regularization $\mathcal{R}(z_i; \hat{y}_i)$, and (iii) Knowledge Bank-guided Consistency Regularization $\mathcal{R}(z_i; \mathcal{B})$, as follows:

$$\mathcal{L}_{\text{online}}(z_i, \boldsymbol{\mu}, \boldsymbol{\Sigma}) = -z_i^\top \log \mathbb{P}_i + \mathcal{R}(z_i; \hat{y}_i) + \mathcal{R}(z_i; \mathcal{B}), \quad (5)$$

$$\text{where } \mathcal{R}(z_i; \hat{y}_i) = \mathrm{KL}(z_i \| \hat{y}_i) + \beta \sum_{k=1}^K \mathrm{KL}\left(\mathcal{N}(\hat{\boldsymbol{\mu}}_k, \hat{\boldsymbol{\Sigma}}_k) \| \mathcal{N}(\boldsymbol{\mu}_k, \boldsymbol{\Sigma})\right), \quad (6)$$

$$\mathcal{R}(z_i; \mathcal{B}) = -\sum_{j \in \mathcal{B}} \hat{y}_j^\top \log \mathbb{P}_j - \sum_{j \in \mathcal{B}} w_{ij} z_i^\top \hat{y}_j. \quad (7)$$

Here, $z_i \in \Delta^K$ denotes the predicted class distribution for test input $\mathbf{x}_i$, and $\mathbb{P}_i$ is the GDA likelihood vector for test sample $\mathbf{x}_i$ defined in Eq. (3). The three terms are explained below:

| **Algorithm 1** ADAPT: Online TTA | **Algorithm 2** ADAPT: Transductive TTA |
|---|---|
| 1: **Input**: Test data $\mathcal{D}_u$, class prototypes $\mathbf{t}$ and knowledge bank size $L$ | 1: **Input**: Test data $\mathcal{D}_u = \{\mathbf{x}_i\}_{i=1}^N$, class prototypes $\mathbf{t}$ and knowledge bank size $L$ |
| 2: **Initialize:** $\hat{\boldsymbol{\mu}} \leftarrow \mathbf{t}$ | 2: **Initialize:** $\hat{\boldsymbol{\mu}} \leftarrow \mathbf{t}$ |
| 3: **for** $\mathbf{x}_i \in \mathcal{D}_u$ **do** | 3: Compute $\text{Conf}(\mathbf{x})$ for all data by Eq. (2) |
| 4:     Compute $\text{Conf}(\mathbf{x}_i)$ by Eq. (2) | 4: **for** $\mathcal{B}_k \in \mathcal{B}$ **do** |
| 5:     Update $\mathcal{B}_k$ with $\mathbf{x}_i$ if high-confidence | 5:     Cache Top-$L$ confidence samples |
| 6:     Update $\boldsymbol{\mu}^*$ and $\boldsymbol{\Sigma}$ by Eq. (9)-(10) | 6: **end for** |
| 7:     Compute $z_i^*$ by Eq. (8) | 7: Update $\boldsymbol{\Sigma}$ and $\boldsymbol{\mu}^*$ by Eq. (10)-(67) |
| 8: **end for** | 8: Compute $z^* = \{z_i^*\}_{i-1}^N$ by Eq. (8) |
| 9: **return** $\{z_i^*\}_{i-1}^N$ | 9: **return** $z^*$ |

- *Online Negative Log-Likelihood*: It encourages $z_i$ to align with the class-conditional Gaussian likelihoods $\mathbb{P}_i$, playing the role of an online maximum likelihood estimation objective. However, due to the streaming nature of test-time inference, early predictions may be unreliable, motivating the need for additional regularization.

- *CLIP Prior-based Regularization*: This term introduces a prior over $z_i$ using the zero-shot CLIP prediction $\hat{y}_i$, enforcing semantic consistency between the predicted label and pretrained vision-language alignment. In addition, it softly constrains the learned GDA parameters $(\boldsymbol{\mu}_k, \boldsymbol{\Sigma})$ to remain close to the CLIP-derived estimates $(\hat{\boldsymbol{\mu}}_k, \hat{\boldsymbol{\Sigma}}_k)$[3], stabilizing adaptation across distribution shifts. The balancing factor $\beta$ controls the strength of this prior.

- *Knowledge Bank-guided Consistency Regularization*: To mitigate the bias introduced by single-sample updates in online scenarios, this term leverages the constructed knowledge banks $\mathcal{B}$ to provide stable, context-aware regularization. Instead of enforcing alignment solely between the current prediction $z_i$ and its estimated likelihood, which may be unreliable due to early-stage noise or distribution shift (see Table 5), we incorporate additional supervision from reliable historical samples. Specifically, the first component enforces consistency between the GDA likelihoods $\mathbb{P}_j$ and their pseudo-labels $\hat{y}_j$ within $\mathcal{B}$. The second component aligns the current prediction $z_i$ with these pseudo-labels, weighted by cosine similarity $w_{ij} = \max(0, f_i^\top f_j)$ to reflect semantic proximity. This memory-based regularization stabilizes learning dynamics, corrects transient errors, and promotes smoother decision boundaries, particularly under shifting data distributions.

**Closed-form Solution without Sub-iterations.** In streaming scenarios, where test samples arrive sequentially and inference must be performed online, iterative optimization (*e.g.*, Majorization-Minimization (MM) algorithms) is prohibitively expensive. As shown in Figure 1, we therefore derive a closed-form label estimator by minimizing the regularized online objective in Eq. (5) that enables one-pass, sub-iteration-free inference as follows (proof in Section A.2):

$$z_{i,k}^* = \frac{\hat{y}_{i,k} \cdot \exp\left(\tilde{y}_{i,k} + \sum_{j \in \mathcal{B}_k} w_{ij} \hat{y}_{j,k}\right)}{\sum_{k=1}^K \hat{y}_{i,k} \cdot \exp\left(\tilde{y}_{i,k} + \sum_{j \in \mathcal{B}_k} w_{ij} \hat{y}_{j,k}\right)}, \tag{8}$$

which fuses three sources of information: CLIP zero-shot logits $\hat{y}_{i,k}$ in Eq. (1), GDA predictions $\tilde{y}_{i,k}$ in Eq. (4), and class-conditional consistency via the knowledge bank.

We estimate the mean vectors by taking the derivative and setting it to zero, which yields (proof in Section A.2): $\boldsymbol{\mu}_k = (z_{i,k}\mathbf{x}_i + \sum_{j \in \mathcal{B}_k} \hat{y}_{j,k}\mathbf{x}_j + \beta\hat{\boldsymbol{\mu}}_k)/(z_{i,k} + \sum_{j \in \mathcal{B}_k} \hat{y}_{j,k} + \beta)$. To enhance robustness against noisy predictions at test time and prevent early overfitting to individual samples, we exclude the current instance $\mathbf{x}_i$ from the class mean $\boldsymbol{\mu}_k$ estimation. Instead, we rely on the accumulated high-confidence features stored in the knowledge bank $\mathcal{B}_k$ and the prior mean $\hat{\boldsymbol{\mu}}_k$. It ensures a stable and adaptive class distribution estimate that strikes a balance between efficiency and accuracy for real-time inference, while avoiding the need for iterative mean and prediction optimization. The final mean vector can be expressed using a scalar $\alpha \in [0, 1]$ (see Section 4.3 for analysis) as:

$$\boldsymbol{\mu}_k^* \leftarrow \alpha\boldsymbol{\mu}_k' + (1 - \alpha)\hat{\boldsymbol{\mu}}_k, \quad \text{where } \boldsymbol{\mu}_k' = \frac{\sum_{j \in \mathcal{B}_k} \hat{y}_{j,k}\mathbf{x}_j}{\sum_{j \in \mathcal{B}_k} \hat{y}_{j,k}}, \alpha = \frac{\sum_{j \in \mathcal{B}_k} \hat{y}_{j,k}}{\sum_{j \in \mathcal{B}_k} \hat{y}_{j,k} + \beta}. \tag{9}$$

---

[3]We define the distribution with $\hat{\boldsymbol{\mu}}$ from class prototypes and $\hat{\boldsymbol{\Sigma}}$ from prompt variance or a fixed diagonal.

In high-dimensional feature spaces, directly inverting the empirical covariance matrix is often ill-conditioned and leads to biased estimates [51]. To address this, we adopt a Bayesian ridge estimator [23] with shrinkage regularization to approximate the shared covariance:

$$\mathbf{\Sigma} = \frac{1}{N_{\mathcal{B}}} \sum_{k=1}^{K} \sum_{j \in \mathcal{B}_k} (\mathbf{x}_j - \boldsymbol{\mu}_k^*)(\mathbf{x}_j - \boldsymbol{\mu}_k^*)^\top, \quad \mathbf{\Sigma}^{-1} = d\left((N_{\mathcal{B}} - 1)\mathbf{\Sigma} + \mathrm{tr}(\mathbf{\Sigma})I_d\right)^{-1}. \tag{10}$$

Where $d$ is the feature dimension, and $N_{\mathcal{B}} \leq K \times L$ is the total number of cached samples in dynamic knowledge banks $\mathcal{B}$. This regularization ensures numerical stability and reliable density estimates without requiring access to source data.

**Extension to the Transductive Optimization.** In the transductive setting, the entire test set $\mathcal{D}_u = \{\mathbf{x}_i\}_{i=1}^N$ is available during inference, allowing us to jointly optimize label distributions over all test instances rather than updating them sequentially. We extend the online regularized objective in Eq. (5) to a transductive objective as follows:

$$\mathcal{L}_{\mathrm{trans}}(z, \boldsymbol{\mu}, \mathbf{\Sigma}) = -\sum_{i=1}^N z_i^\top \log \mathbb{P}_i + \sum_{i=1}^N \mathcal{R}(z_i; \hat{y}_i) + \sum_{i=1}^N \mathcal{R}(z_i; \mathcal{B}). \tag{11}$$

Similar to the online case, the optimization in Eq. (11) is convex with respect to $z$ and admits a closed-form label estimator. Because each sample remains conditionally independent, the form of the solution is unchanged from the online case in Eq. (8) (proof in Section A.3).

Taking the derivative of Eq. (11) with respect to $\boldsymbol{\mu}_k$ and setting it to zero yields the following solution (proof in Section A.3): $\boldsymbol{\mu}_k = (\sum_{i=1}^N z_{i,k}\mathbf{x}_i + \sum_{j \in \mathcal{B}_k} \hat{y}_{j,k}\mathbf{x}_j + \beta\hat{\boldsymbol{\mu}}_k)/(\sum_{i=1}^N z_{i,k} + \sum_{j \in \mathcal{B}_k} \hat{y}_{j,k} + \beta)$. In practice, to further improve efficiency and avoid the iterative updates required by MM-like algorithms, we substitute $z_i$ with the CLIP zero-shot predictions $\hat{y}_i$ in Eq. (1), yielding a one-pass estimate for class means $\boldsymbol{\mu}_k$:

$$\boldsymbol{\mu}_k^* \leftarrow \alpha\boldsymbol{\mu}_k' + (1-\alpha)\hat{\boldsymbol{\mu}}_k, \boldsymbol{\mu}_k' = \frac{\sum_{i=1}^N \hat{y}_{i,k}\mathbf{x}_i + \sum_{j \in \mathcal{B}_k} \hat{y}_{j,k}\mathbf{x}_j}{\sum_{i=1}^N \hat{y}_{i,k} + \sum_{j \in \mathcal{B}_k} \hat{y}_{j,k}}, \alpha = \frac{\sum_{i=1}^N \hat{y}_{i,k} + \sum_{j \in \mathcal{B}_k} \hat{y}_{j,k}}{\sum_{i=1}^N \hat{y}_{i,k} + \sum_{j \in \mathcal{B}_k} \hat{y}_{j,k} + \beta}. \tag{12}$$

We apply the same covariance matrix estimation strategy as in the online setting (Eq. 10). We summarize the procedure for online test-time adaptation in Algorithm 1, and present the overall transductive adaptation process in Algorithm 2. Notably, both procedures are fully optimization-free, relying on one-pass closed-form updates for efficient adaptation.

**Remark.** To enable efficient and stable adaptation in streaming settings, we exclude the current test sample $\mathbf{x}_i$ from updating class means. This prevents noise from low- or mid-confidence predictions and avoids error propagation during early stages. Only high-confidence samples are stored in the knowledge bank for future updates. As shown in Table 8, this achieves similar performance with reduced computation. In the transductive setting, the class means $\boldsymbol{\mu}_k$ are estimated using the full unlabeled test set and accumulated confident samples. Instead of optimizing latent labels, we apply CLIP's soft predictions $\hat{y}_i$ for one-pass, closed-form estimation. This avoids iterative overhead while leveraging globally consistent signals. Aggregated soft labels offer a reliable proxy, matching or outperforming iterative methods in both accuracy and stability (see more details in Section A.4).

## 4 Experiments

### 4.1 Setup

**Dataset.** We evaluated the proposed ADAPT on three different tasks: natural distribution shift, fine-grained categorization, and corruption robustness. Specifically, for natural distribution shift, we use multiple datasets including ImageNet [4], ImageNet-A [17], ImageNet-R [15],ImageNet-V [40], and ImageNet-Sketch [50]. The corruption robustness task is evaluted on ImageNet-C [16], which contains 15 corruption types covering noise, blur, weather, and digital artifacts. We also evaluate performance on 10 fine-grained recognition datasets: Aircraft [32], Caltech101 [8], Cars [22], DTD [3], EuroSAT [14], Flower102 [36], Food101 [1], Pets [37], SUN397 [53] and UCF101 [45].

Table 2: Top-1 accuracy (%) comparison on natural distribution shift.

| | Method | BP-free | ImageNet | ImageNet-A | ImageNet-V | ImageNet-R | ImageNet-S | OOD Avg. | Avg. |
|---|---|---|---|---|---|---|---|---|---|
| Online | CLIP [39] | - | 66.74 | 47.79 | 60.89 | 73.99 | 46.12 | 57.20 | 59.11 |
| | Tip-Adapter [63] | ✗ | 70.75 | 51.04 | 63.41 | 77.76 | 48.88 | 60.27 | 62.37 |
| | TPT [33] | ✗ | 68.98 | 54.77 | 63.45 | 77.06 | 47.97 | 60.81 | 62.45 |
| | DiffTPT [9] | ✗ | 70.30 | 55.68 | 65.10 | 75.00 | 46.80 | 60.65 | 62.58 |
| | C-TPT [55] | ✗ | 68.50 | 51.60 | 62.70 | 76.00 | 47.90 | 59.55 | 61.34 |
| | DMN [65] | ✗ | 72.25 | 58.28 | 65.17 | 78.55 | **53.20** | 63.80 | 65.49 |
| | DPE [61] | ✗ | **71.91** | 59.63 | **65.44** | 80.40 | 52.26 | 64.43 | 65.93 |
| | TPS [46] | ✗ | 70.38 | 59.21 | 63.80 | 77.49 | 49.57 | 62.52 | 64.09 |
| | DynaPrompt [54] | ✗ | 69.61 | 56.17 | 64.67 | 78.17 | 48.22 | 61.81 | 63.37 |
| | B²TPT [34] | ✗ | 69.57 | 55.26 | 65.40 | 78.64 | 49.53 | 62.21 | 63.68 |
| | MTA [57] | ✓ | 70.08 | 58.06 | 64.24 | 78.33 | 49.61 | 62.56 | 64.06 |
| | TDA [21] | ✓ | 69.51 | 60.11 | 64.67 | 80.24 | 50.54 | 63.89 | 65.01 |
| | ZERO [7] | ✓ | 69.31 | 59.61 | 64.16 | 77.22 | 48.40 | 62.35 | 63.74 |
| | AWT [70] | ✓ | 71.32 | 60.33 | 65.15 | 80.64 | 51.60 | 64.43 | 65.81 |
| | RA-TTA [24] | ✓ | 70.58 | 59.21 | 64.16 | 79.68 | 50.83 | 63.47 | 64.89 |
| | BCA [67] | ✓ | 70.22 | 61.14 | 64.90 | 80.72 | 50.87 | 64.41 | 65.57 |
| | TCA [52] | ✓ | 68.88 | 50.13 | 62.10 | 77.11 | 48.95 | 59.57 | 61.43 |
| | Dota [12] | ✓ | 70.68 | 61.19 | 64.41 | **81.17** | 51.33 | 64.53 | 65.76 |
| | ADAPT | ✓ | 70.91 | **63.32** | 64.64 | 80.66 | 53.13 | **65.44** | **66.53** |
| Trans. | GDA-CLIP [51] | ✓ | 64.13 | 19.72 | 55.67 | 55.30 | 34.32 | 41.25 | 45.83 |
| | TransCLIP [59] | ✓ | 70.30 | 49.50 | 62.30 | 75.00 | 49.70 | 59.13 | 61.36 |
| | Frolic [69] | ✓ | 70.90 | 60.40 | 64.70 | **80.70** | 53.30 | 64.78 | 66.00 |
| | TIMO [28] | ✓ | 64.63 | 22.06 | 56.40 | 58.47 | 35.96 | 43.22 | 47.50 |
| | ADAPT | ✓ | **71.56** | **63.77** | **65.59** | 80.64 | **53.87** | **65.97** | **67.09** |

Table 3: Top-1 accuracy (%) comparison on corruption robustness.

| | Method | Blur | | | | Weather | | | | Digital | | | | Noise | | | Avg. |
|---|---|---|---|---|---|---|---|---|---|---|---|---|---|---|---|---|---|
| | | Defo. | Glas. | Moti. | Zoom | Snow | Fros. | Fog | Brig. | Cont. | Elas. | Pix. | JPEG | Gauss. | Shot | Impu. | |
| Online | CLIP [39] | 24.25 | 15.71 | 24.46 | 22.60 | 33.08 | 31.06 | 37.61 | 55.62 | 17.11 | 13.43 | 33.04 | 33.70 | 13.25 | 14.16 | 13.48 | 25.50 |
| | TPT [33] | **27.56** | 15.48 | 26.16 | **26.94** | **36.74** | 34.28 | 39.38 | 60.22 | 16.96 | 15.64 | **40.74** | **37.90** | 10.64 | 11.94 | 10.92 | 27.43 |
| | DiffTPT [9] | 25.63 | 16.96 | 26.74 | 25.40 | 35.99 | 34.57 | 39.83 | 59.01 | 17.32 | **17.16** | 38.43 | 35.47 | 12.97 | 13.60 | 13.21 | 27.49 |
| | TDA [21] | 26.53 | 17.91 | **27.35** | 25.90 | 36.50 | **34.84** | 40.53 | 58.57 | **20.16** | 16.62 | 35.65 | 36.69 | 15.42 | 16.46 | **16.03** | 28.34 |
| | DMN [65] | 26.06 | 17.19 | 26.61 | 25.23 | 34.81 | 33.48 | 38.93 | 58.70 | 19.38 | 15.40 | 35.32 | 36.49 | 14.33 | 15.33 | 14.69 | 27.46 |
| | ADAPT | 26.30 | **18.01** | 27.31 | 25.54 | 36.19 | 34.67 | **40.96** | 60.29 | 19.95 | 16.09 | 37.44 | 37.22 | **15.76** | **16.84** | 15.90 | **28.56** |
| Trans. | ZLaP [20] | 24.88 | 16.13 | 25.77 | 24.36 | 34.43 | 32.63 | 38.56 | 58.42 | 17.53 | 14.21 | 33.72 | 35.52 | 12.83 | 14.03 | 13.27 | 26.42 |
| | TransCLIP [59] | 25.35 | 16.40 | 25.53 | 23.22 | 34.58 | 32.47 | 39.65 | 59.04 | 17.72 | 14.76 | 35.22 | 35.53 | 14.82 | 16.11 | 15.60 | 27.07 |
| | StatA [58] | 20.23 | 13.29 | 20.38 | 18.84 | 31.30 | 29.80 | 34.58 | 54.79 | 11.24 | 11.80 | 26.31 | 33.20 | 9.58 | 10.52 | 10.12 | 22.40 |
| | ADAPT | **27.98** | **19.78** | **29.00** | **27.38** | **38.09** | **36.44** | **42.43** | **62.21** | **21.94** | **18.40** | **39.89** | **38.23** | **17.71** | **18.81** | **18.09** | **30.29** |

**Evaluation Protocol and Implementation Details.** We evaluate ADAPT under two TTA protocols: Online (batch size=1) and Transductive. In the online setting, the model processes one sample at a time with access to current and past inputs, following the sequential setup in [33, 21]. In the transductive setting [59, 60], all test samples are accessible at once. We use CLIP [39] ViT-B/16 as the backbone. We set the coefficient $\alpha$ to 0.9, and assign the knowledge bank size $L$ as 16 for online and 6 for transductive evaluation. All experiments are conducted on an NVIDIA RTX A6000 GPU.

## 4.2 Main Results

**Task 1: Natural Distribution Shift.** As shown in Table 2, ADAPT achieves the highest average accuracy of 66.53% in the online setting, outperforming all optimization-based and backpropagation-free baselines. It also surpasses strong transductive methods like TransCLIP and Frolic, despite not accessing the full target set. In the transductive setting, ADAPT further improves to 67.09%, ranking first on three out of four OOD benchmarks and showing consistent robustness under domain shifts.

**Task 2: Corruption Robustness.** Table 3 reports results under a variety of synthetic corruptions. ADAPT achieves the best performance with 28.56% accuracy in the online setting and 30.29% in the transductive setting, consistently surpassing all baselines, including transductive methods. These results show the robustness and adaptability of ADAPT even under severe visual degradations.

**Task 3: Fine-Grained Categorization.** As shown in Table 4, ADAPT outperforms the CLIP zero-shot baseline by 7.31% (online) and 8.98% (transductive), demonstrating strong generalization to fine-grained variations. It achieves the highest overall accuracy among backpropagation-free methods without labels or gradients, and remains competitive with transductive approaches using full target data. Despite using neither source data nor external supervision, ADAPT narrows the gap to Oracle performance within 5%, which is obtained by estimating target distributions using ground-truth labels and full access to the test set, offering a unified and practical solution for real-world TTA.

Table 4: Top-1 accuracy (%) comparison on fine-grained categorization.

| | Method | BP-free | Aircraft | Caltech | Cars | DTD | EuroSAT | Flower | Food101 | Pets | Sun397 | UCF101 | **Avg.** |
|---|---|---|---|---|---|---|---|---|---|---|---|---|---|
| **Online** | CLIP [39] | - | 23.70 | 92.98 | 65.24 | 44.44 | 41.42 | 67.28 | 83.80 | 87.98 | 62.55 | 65.08 | 63.45 |
| | TPT [33] | ✗ | 24.78 | 94.16 | 66.87 | 47.75 | 42.44 | 68.98 | 84.67 | 87.79 | 65.50 | 68.04 | 65.10 |
| | DiffTPT [9] | ✗ | 25.60 | 92.49 | 67.01 | 47.00 | 43.13 | 70.10 | 87.23 | 88.22 | 65.74 | 68.22 | 65.47 |
| | C-TPT [55] | ✗ | 24.00 | 93.60 | 65.80 | 46.00 | 43.20 | **79.80** | 83.70 | 88.20 | 64.80 | 65.70 | 64.48 |
| | DMN [65] | ✗ | **30.03** | **95.38** | 67.96 | **55.85** | 59.43 | 74.49 | 85.08 | **92.04** | 70.18 | **72.51** | 70.30 |
| | TPS [29] | ✗ | 26.27 | 94.56 | 67.00 | 53.80 | 42.11 | 71.69 | 84.78 | 87.82 | 68.25 | 71.18 | 66.75 |
| | DPE [61] | ✗ | 28.95 | 94.81 | 67.31 | 54.20 | 55.79 | 75.07 | 86.17 | 91.14 | 70.07 | 70.44 | 69.40 |
| | HisTPT [62] | ✗ | 26.90 | 94.50 | 69.20 | 48.90 | 49.70 | 71.20 | **89.30** | 89.10 | 67.20 | 70.10 | 67.61 |
| | DynaPrompt [54] | ✗ | 24.33 | 94.32 | 67.65 | 47.96 | 42.28 | 69.95 | 85.42 | 88.28 | 66.32 | 68.72 | 65.52 |
| | MTA [57] | ✓ | 25.32 | 94.13 | 66.36 | 45.59 | 38.71 | 68.26 | 84.95 | 88.22 | 64.98 | 68.11 | 64.46 |
| | TDA [21] | ✓ | 23.91 | 94.24 | 67.28 | 47.40 | 58.00 | 71.42 | 86.14 | 88.63 | 67.62 | 70.66 | 67.53 |
| | ZLaP [20] | ✓ | 25.40 | 93.10 | 65.60 | 48.60 | 55.60 | 73.50 | 86.90 | 87.10 | 67.40 | 71.50 | 67.47 |
| | ZERO [7] | ✓ | 25.21 | 93.66 | 68.04 | 46.12 | 34.33 | 67.68 | 86.53 | 87.75 | 65.03 | 67.77 | 64.21 |
| | BCA [67] | ✓ | 28.59 | 94.69 | 66.86 | 53.49 | 56.63 | 73.12 | 85.97 | 90.43 | 68.41 | 67.59 | 68.58 |
| | OGA [10] | ✓ | 23.20 | 93.60 | 68.10 | 47.90 | 54.20 | 69.20 | 85.60 | 89.40 | 67.90 | 71.40 | 67.05 |
| | TCA [52] | ✓ | 24.87 | 93.63 | 65.33 | 46.16 | **70.43** | 73.33 | 85.31 | 89.53 | 65.92 | 72.38 | 68.69 |
| | Dota [12] | ✓ | 25.59 | 94.32 | **69.48** | 47.87 | 57.65 | 74.67 | 87.02 | 91.69 | 69.70 | 72.06 | 69.01 |
| | ADAPT | ✓ | 28.95 | 94.48 | 68.19 | 55.20 | 68.19 | 75.56 | 83.81 | 92.01 | **70.57** | 70.66 | **70.76** |
| **Trans.** | GDA-CLIP [51] | ✓ | 18.69 | 87.53 | 60.78 | 46.81 | 49.92 | 72.65 | 78.25 | 89.90 | 63.60 | 68.70 | 63.68 |
| | ZLaP [20] | ✓ | 26.30 | 91.80 | 66.80 | 46.00 | 57.70 | 67.90 | **87.20** | 87.90 | 67.80 | 73.80 | 67.32 |
| | TransCLIP [59] | ✓ | 26.90 | 92.70 | 69.40 | 49.50 | 65.10 | 76.70 | 87.10 | 92.60 | 68.90 | 74.40 | 70.33 |
| | Frolic [69] | ✓ | **31.40** | 95.10 | 69.10 | 56.10 | 58.50 | 74.80 | 87.10 | **92.90** | 70.80 | **75.20** | 71.10 |
| | StatA [58] | ✓ | 24.70 | 94.20 | 68.00 | 48.40 | **67.30** | 75.20 | 87.10 | 92.40 | 68.70 | 73.50 | 69.95 |
| | ADAPT | ✓ | 30.81 | **95.46** | **71.32** | **56.86** | 65.93 | **80.11** | 85.15 | 92.59 | **72.25** | 73.86 | **72.43** |
| | Oracle ADAPT | ✓ | 41.88 | 98.26 | 82.89 | 60.87 | 56.51 | 81.93 | 85.74 | 92.61 | 80.04 | 90.14 | 77.09 |

Table 5: Ablation study on components.

| $\mathcal{B}$ | Update $\mu$ | Update $\Sigma$ | Task 1 | Task 2 | Task 3 |
|---|---|---|---|---|---|
| ✗ | ✗ | ✗ | 59.11 | 25.50 | 63.45 |
| ✗ | ✗ | ✓ | 49.64 | 9.58 | 60.02 |
| ✗ | ✓ | ✗ | 61.54 | 25.42 | 67.03 |
| ✗ | ✓ | ✓ | 49.65 | 9.58 | 60.04 |
| ✓ | ✗ | ✗ | 64.89 | 25.08 | 67.06 |
| ✓ | ✗ | ✓ | 64.05 | 19.49 | 67.95 |
| ✓ | ✓ | ✗ | 65.27 | 25.67 | 67.43 |
| ✓ | ✓ | ✓ | **66.53** | **28.56** | **70.76** |

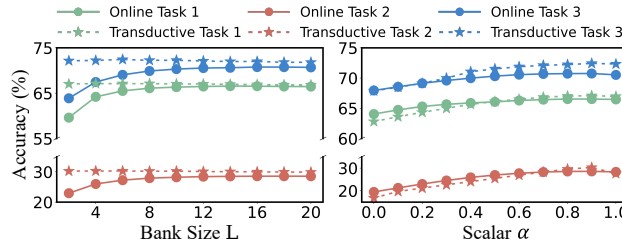

Figure 2: Hyperparameter analysis.

## 4.3 Ablation Studies and Further Analysis

**Ablation Study.** We ablate the key components in ADAPT. As shown in Table 5, removing the knowledge bank $\mathcal{B}$ causes notable performance degradation, especially when updating $\Sigma$, due to noisy statistics from test-time predictions. This confirms that our bank-guided regularization in Eq. (7) effectively mitigates likelihood bias in the online setting. With $\mathcal{B}$, adapting either $\mu$ or $\Sigma$ provides moderate gains, while jointly updating both yields the best results. These findings highlight the complementary roles of adaptive statistics and memory-guided estimation.

**Hyperparameter Analysis.** We analyze the impact of the knowledge bank size $L$ and the coefficient $\alpha$ in Eq.(9) and Eq.(67), where $\alpha$ controls the balance between the CLIP prior and the empirical mean of confident test features. Figure 2 (left) shows that online performance is unstable when $L$ is small due to limited statistics, but stabilizes once $L \geq 8$. In contrast, the transductive setting is robust to $L$ as all target data are available. We use $L=16$ for online and $L=6$ for transductive. As shown in Figure 2 (right), performance steadily improves with larger $\alpha$, confirming that ADAPT benefits from reliable historical samples over static CLIP priors. We set $\alpha=0.9$ to balance prior knowledge and adaptivity.

**Cost Comparison.** In Table 6, we evaluate all methods under the same protocol and computational resource (NVIDIA RTX A6000) to ensure a fair comparison. In the online setting, ADAPT achieves the highest accuracy with just 1h 11m and 0.93GB, notably lower than TPT. In the transductive setting, it builds the knowledge bank once and runs in a single pass, requiring only 0.73 minutes and 3.37GB. These results highlight ADAPT as a practical solution balancing accuracy and efficiency.

**Comparison with Different Mean Initialization $\hat{\mu}$.** We study how different initializations of the mean $\hat{\mu}$ in Eq. (9) affect performance. We compare 6 types of text-derived prototypes: CLIP's vanilla template, 7 generic templates, 80 ImageNet-specific templates, GPT-generated descriptions in [70], and its variants. As shown in Table 7, diverse and descriptive initializations improve performance, though ADAPT remains robust across all settings. We use GPT-based descriptions in our main results.

Table 6: Efficiency comparison on ImageNet.

| | Method | BP-free | Acc (%) ↑ | Gain (%) ↑ | Time ↓ | Mem.(GB) ↓ |
|---|---|---|---|---|---|---|
| Online | CLIP [39] | ✓ | 66.74 | - | 8m | 0.79 |
| | TPT [33] | ✗ | 68.95 | 2.21 | 9h 45m | 4.29 |
| | DiffTPT [9] | ✗ | 70.30 | 3.56 | > 20h | 4.60 |
| | TDA [21] | ✓ | 69.51 | 2.77 | 50m | 0.84 |
| | TPS [46] | ✗ | 70.38 | 3.64 | 1h 19m | 1.71 |
| | ADAPT | ✓ | 70.91 | 4.17 | 1h 11m | 0.93 |
| Trans. | GDA-CLIP [51] | ✓ | 64.13 | -2.61 | 1.31m | 10.03 |
| | TransCLIP [59] | ✓ | 70.30 | 3.56 | 1.34m | 16.17 |
| | StatA [58] | ✓ | 69.90 | 3.16 | 1.5m | 20.74 |
| | ADAPT | ✓ | 71.56 | 4.82 | 0.73m | 3.37 |

Table 7: Mean initialization comparison.

| Mean initialization $\hat{\mu}$ | Task 1 | Task 2 | Task 3 |
|---|---|---|---|
| Vanilla [39] | 64.70 | 27.52 | 67.43 |
| Ensemble [39] | 66.56 | 28.54 | 67.74 |
| CLIP Template [39] | 66.51 | 28.44 | 67.62 |
| GPT [70] | 66.53 | 28.56 | 70.76 |
| GPT & Ensemble | 66.57 | 28.98 | 69.95 |
| GPT & CLIP Template | 66.58 | 28.91 | 69.90 |

Table 8: Closed-form vs Iterative optimization.

| | Sub-iter. | Task 1 | Task 2 | Task 3 | Time ↓ |
|---|---|---|---|---|---|
| Online | Iterative | 66.74 | 28.68 | 71.05 | 4h 32m |
| | Closed | 66.53 | 28.56 | 70.76 | 1h 11m |
| Trans. | Iterative | 66.88 | 27.95 | 73.98 | 0.89m |
| | Closed | 67.09 | 30.29 | 72.43 | 0.73m |

Table 9: Effect of test-time input order in the online adaptation setting.

| Order | Task 1 | Task 2 | Task 3 | Time ↓ |
|---|---|---|---|---|
| Hard-to-Easy | 65.37 | 27.29 | 68.29 | 1h 57m |
| Easy-to-Hard | 66.79 | 29.09 | 70.97 | 30m |
| Random (Ours) | 66.53 | 28.56 | 70.76 | 1h 11m |

Figure 3: Visualization of decision boundaries on ImageNet-A. The colors indicate different classes.

**Comparison with Iterative Optimization.** To avoid costly iterative optimization, we proposed a closed-form, one-pass approximation as in Eq. (9) and Eq. (67). We evaluate it across three tasks in both settings. As shown in Table 8, our closed-form solution matches the accuracy of iterative methods while achieving a 4× speedup on ImageNet in the online scenario. The transductive results further confirm that our method offers an optimal trade-off between efficiency and effectiveness.

**Effects of Online Input Order.** We study how test-time input order affects online adaptation, comparing random (default), hard-to-easy, and easy-to-hard confidence (Eq. (2)) orderings. As shown in Table 9, easy-to-hard ordering performs best, as early high-confidence samples help build reliable class statistics. Despite this, our default random order still yields competitive results, showing the robustness and practicality of ADAPT without relying on input scheduling.

**Visualization.** Figure 3 illustrates the decision boundaries of different TTA methods. CLIP, TPT, and TDA rely on feature similarity, lacking explicit class distribution modeling. Their boundaries are thus irregular and unstable near class overlaps, leading to high intra-class variance and poor separability. In contrast, our ADAPT builds a Gaussian-based model that explicitly estimates the class distribution without training, yielding more compact clusters and smoother boundaries.

## 5 Limitation and Conclusion

**Limitation.** Our method provides closed-form, backpropagation-free updates, making it suitable for scenarios with low latency or limited resources. This efficiency comes from modeling class-conditional features using a Gaussian assumption with shared covariance. While this strategy performs well across diverse benchmarks, it may struggle to capture complex or multi-modal structures in challenging real-world data. Exploring more flexible distribution models, such as Gaussian mixtures or non-parametric estimators, without compromising efficiency remains a promising direction.

**Conclusion.** We propose ADAPT, a distribution-aware and backpropagation-free TTA framework. Unlike prior methods that rely on gradient updates or iterative optimization, ADAPT performs efficient, training-free adaptation by estimating class-conditional distributions from test-time features using closed-form updates. Extensive experiments show that ADAPT is robust and scalable, making it suitable for deployment in dynamic or resource-constrained settings.

**Acknowledgement.** This work was partly supported by Institute of Information & communications Technology Planning & Evaluation (IITP) grant funded by the Korea government(MSIT) (No.RS-2025-02217960, Development of a human-oriented AI model that imitates the human growth process based cognitive science). This research was partly supported by the MSIT(Ministry of Science, ICT), Korea, under the Global Research Support Program in the Digital Field program(RS-2024-00425354) supervised by the IITP(Institute for Information & Communications Technology Planning & Evaluation). This research was partly supported by the MSIT(Ministry of Science and ICT), Korea, under the Graduate School of Metaverse Convergence support program(IITP-2025-RS-2023-00254129) supervised by the IITP(Institute for Information & Communications Technology Planning & Evaluation).

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

## Technical Appendices and Supplementary Material

This appendix provides a detailed theoretical analysis of our method, along with additional experimental results. The contents are organized as follows:

- **Appendix A: Theoretical Analysis**

- **Appendix B: Additional Experimental Results**

## A  Theoretical Analysis

### A.1  Proof of Gaussian Discriminant Analysis

This subsection provides the Gaussian Discriminant Analysis (GDA) decision rule under class-conditional Gaussian assumptions and serves as the theoretical justification for Eq. (4) in the main paper. Specifically, we derive the discriminant function of GDA under the assumption that class-conditional distributions follow multivariate Gaussians with a shared covariance matrix:

$$\mathbb{P}(\mathbf{x}|y_k) = \mathcal{N}(\mathbf{x}; \boldsymbol{\mu}_k, \boldsymbol{\Sigma}) = \frac{1}{\sqrt{(2\pi)^d |\boldsymbol{\Sigma}|}} \exp\left(-\frac{1}{2}(\mathbf{x} - \boldsymbol{\mu}_k)^\top \boldsymbol{\Sigma}^{-1}(\mathbf{x} - \boldsymbol{\mu}_k)\right). \tag{13}$$

Using Bayes' rule, the posterior probability can be expressed as: $\mathbb{P}(y_k|\mathbf{x}) = \frac{\mathbb{P}(\mathbf{x}|y_k)\,\mathbb{P}(y_k)}{\mathbb{P}(\mathbf{x})} \propto \pi_k \mathcal{N}(\mathbf{x}; \boldsymbol{\mu}_k, \boldsymbol{\Sigma})$, which reformulates the $K$-way classification problem into a maximum likelihood estimation task. Considering that the target source domain generally is class-balanced, we set the prior class distribution to be uniform: $\mathbb{P}(y_k) = \frac{1}{K}$. Then, we can derive the GDA classifier by maximizing the likelihood as following:

$$\log \mathcal{N}(\mathbf{x}; \boldsymbol{\mu}_k, \boldsymbol{\Sigma}) \cdot \pi_k = \log \frac{1}{\sqrt{(2\pi)^d |\boldsymbol{\Sigma}|}} \exp\left(-\frac{1}{2}(\mathbf{x} - \boldsymbol{\mu}_k)^\top \boldsymbol{\Sigma}^{-1}(\mathbf{x} - \boldsymbol{\mu}_k)\right) + \log \pi_k \tag{14}$$

$$= -\frac{d}{2}\log 2\pi - \frac{1}{2}\log|\boldsymbol{\Sigma}| - \frac{1}{2}(\mathbf{x} - \boldsymbol{\mu}_k)^\top \boldsymbol{\Sigma}^{-1}(\mathbf{x} - \boldsymbol{\mu}_k) + \log \pi_k \tag{15}$$

$$= -\frac{d}{2}\log 2\pi - \frac{1}{2}\log|\boldsymbol{\Sigma}| - \frac{1}{2}\mathbf{x}^\top \boldsymbol{\Sigma}^{-1}\mathbf{x} + \boldsymbol{\mu}_k^\top \boldsymbol{\Sigma}^{-1}\mathbf{x} \tag{16}$$

$$\quad - \frac{1}{2}\boldsymbol{\mu}_k^\top \boldsymbol{\Sigma}^{-1}\boldsymbol{\mu}_k + \log \pi_k \tag{17}$$

$$= \boldsymbol{\mu}_k^\top \boldsymbol{\Sigma}^{-1}\mathbf{x} - \frac{1}{2}\boldsymbol{\mu}_k^\top \boldsymbol{\Sigma}^{-1}\boldsymbol{\mu}_k + \mathrm{C} \tag{18}$$

$$\stackrel{(a)}{=} \boldsymbol{\mu}_k^\top \boldsymbol{\Sigma}^{-1}\mathbf{x} - \frac{1}{2}\boldsymbol{\mu}_k^\top \boldsymbol{\Sigma}^{-1}\boldsymbol{\mu}_k \tag{19}$$

$$= \mathbf{w}_k^\top \mathbf{x} + b_k. \tag{20}$$

$\stackrel{(a)}{=}$ holds as we can remove all constant terms $\mathrm{C} = \log \pi_k - \frac{d}{2}\log 2\pi - \frac{1}{2}\log|\boldsymbol{\Sigma}| - \frac{1}{2}\mathbf{x}^\top \boldsymbol{\Sigma}^{-1}\mathbf{x}$. Then, the GDA prediction can be expressed as: $\tilde{y}_{i,k} = \mathbf{w}_k^\top \mathbf{x}_i + b_k$, where $\mathbf{w}_k = \boldsymbol{\Sigma}^{-1}\boldsymbol{\mu}_k$, $b_k = -\frac{1}{2}\boldsymbol{\mu}_k^\top \boldsymbol{\Sigma}^{-1}\boldsymbol{\mu}_k$.

## A.2 Online Test-time Distribution Estimation

While GDA offers a principled framework for label estimation using class-conditional statistics, its direct application in the online setting may result in biased likelihood estimates particularly in early stages where data is scarce and predictions tend to be overconfident. To address this, we construct a regularized objective (Eq. (5) in our main paper) that integrates three complementary sources of information: (i) the online negative log-likelihood, $-z_i^\top \log \mathbb{P}_i$; (ii) a prior regularization term based on CLIP zero-shot predictions, $\mathcal{R}(z_i; \hat{y}_i)$; and (iii) a consistency regularization term guided by the knowledge bank, $\mathcal{R}(z_i; \mathcal{B})$. We formulate the full objective as follows:

$$\mathcal{L}_{\text{online}}(z_i, \boldsymbol{\mu}, \boldsymbol{\Sigma}) = -z_i^\top \log \mathbb{P}_i + \mathcal{R}(z_i; \hat{y}_i) + \mathcal{R}(z_i; \mathcal{B}) \tag{21}$$

$$= -z_i^\top \log \mathbb{P}_i + \text{KL}(z_i \| \hat{y}_i) + \beta \sum_{k=1}^{K} \text{KL}\left(\mathcal{N}(\hat{\boldsymbol{\mu}}_k, \hat{\boldsymbol{\Sigma}}_k) \| \mathcal{N}(\boldsymbol{\mu}_k, \boldsymbol{\Sigma})\right) \tag{22}$$

$$- \sum_{j \in \mathcal{B}} \hat{y}_j^\top \log \mathbb{P}_j - \sum_{j \in \mathcal{B}} w_{ij} z_i^\top \hat{y}_j. \tag{23}$$

For the third term, we assume that $x$ follows a Gaussian distribution $\mathcal{N}_0 = \mathcal{N}(\hat{\mu}_k, \hat{\Sigma}_k)$. According Matrix Cookbook [38] and [58], the following identity holds: $\mathbb{E}_{\mathcal{N}_0}\left[(x - \mu_k)^\top \Sigma^{-1}(x - \mu_k)\right] = (\hat{\mu}_k - \mu_k)^\top \Sigma^{-1}(\hat{\mu}_k - \mu_k) + \text{Tr}(\Sigma^{-1}\hat{\Sigma}_k)$. Based on this, the KL divergence between the distributions $\mathcal{N}_0 = \mathcal{N}(\hat{\mu}_k, \hat{\Sigma}_k)$ and $\mathcal{N}_1 = \mathcal{N}(\mu_k, \Sigma))$ is given as:

$$\text{KL}(\mathcal{N}(\hat{\mu}_k, \hat{\Sigma}_k) \| \mathcal{N}(\mu_k, \Sigma)) = \int_x \mathcal{N}_0(x) \log \frac{\mathcal{N}_0(x)}{\mathcal{N}_1(x)} dx = \mathbb{E}_{\mathcal{N}_0}[\log \mathcal{N}_0(x) - \log \mathcal{N}_1(x)] \tag{24}$$

$$= \mathbb{E}_{\mathcal{N}_0}\left[\frac{1}{2} \log \frac{|\Sigma|}{|\hat{\Sigma}_k|} - \frac{1}{2}(x - \hat{\mu}_k)^\top \hat{\Sigma}_k^{-1}(x - \hat{\mu}_k) + \frac{1}{2}(x - \mu_k)^\top \Sigma^{-1}(x - \mu_k)\right] \tag{25}$$

$$= \frac{1}{2}\left(\log \frac{|\Sigma|}{|\hat{\Sigma}_k|} - \mathbb{E}_{\mathcal{N}_0}\left[(x - \hat{\mu}_k)^\top \hat{\Sigma}_k^{-1}(x - \hat{\mu}_k)\right] + \mathbb{E}_{\mathcal{N}_0}\left[(x - \mu_k)^\top \Sigma^{-1}(x - \mu_k)\right]\right) \tag{26}$$

$$\stackrel{(b)}{=} \frac{1}{2}\left(\log \frac{|\Sigma|}{|\hat{\Sigma}_k|} - d + (\hat{\mu}_k - \mu_k)^\top \Sigma^{-1}(\hat{\mu}_k - \mu_k) + \text{Tr}(\Sigma^{-1}\hat{\Sigma}_k)\right), \tag{27}$$

where $d$ is the feature dimension and $\stackrel{(b)}{=}$ holds as:

$$\mathbb{E}_{\mathcal{N}_0}\left[(x - \hat{\mu}_k)^\top \hat{\Sigma}_k^{-1}(x - \hat{\mu}_k)\right] = \mathbb{E}_{\mathcal{N}_0}\left[\text{Tr}\left((x - \hat{\mu}_k)(x - \hat{\mu}_k)^\top \hat{\Sigma}_k^{-1}\right)\right] \tag{28}$$

$$= \text{Tr}(\mathbb{E}_{\mathcal{N}_0}\left[(x - \hat{\mu}_k)(x - \hat{\mu}_k)^\top\right] \hat{\Sigma}_k^{-1}) \tag{29}$$

$$= \text{Tr}(\hat{\Sigma}_k \hat{\Sigma}_k^{-1}) \tag{30}$$

$$= \text{Tr}(\mathbb{I}_K) \tag{31}$$

$$= d \tag{32}$$

### A.2.1 Estimation of Class Mean

To enable efficient distribution estimation in streaming scenarios where test samples arrive sequentially, we estimate the class means $\boldsymbol{\mu}_k$ by taking the derivative of $\mathcal{L}_{\text{online}}$ w.r.t. $\boldsymbol{\mu}_k$:

$$\frac{\partial \mathcal{L}_{\text{online}}}{\partial \boldsymbol{\mu}_k} = -\frac{\partial}{\partial \boldsymbol{\mu}_k}(z_i^\top \log \mathbb{P}_i) + \frac{\partial}{\partial \boldsymbol{\mu}_k}(\beta \sum_{k=1}^K \text{KL}(\mathcal{N}(\hat{\boldsymbol{\mu}}_k, \hat{\boldsymbol{\Sigma}}_k) \| \mathcal{N}(\boldsymbol{\mu}_k, \boldsymbol{\Sigma}))) \tag{33}$$

$$-\frac{\partial}{\partial \boldsymbol{\mu}_k}(\sum_{j \in \mathcal{B}} \hat{y}_j^\top \log \mathbb{P}_j) \tag{34}$$

$$= -\frac{\partial}{\partial \boldsymbol{\mu}_k}(z_i \log(\frac{1}{\sqrt{(2\pi)^d |\boldsymbol{\Sigma}|}} \exp(-\frac{1}{2}(\mathbf{x}_i - \boldsymbol{\mu}_k)^\top \boldsymbol{\Sigma}^{-1}(\mathbf{x}_i - \boldsymbol{\mu}_k)))) \tag{35}$$

$$+\frac{\partial}{\partial \boldsymbol{\mu}_k}(\beta \text{KL}(\mathcal{N}(\hat{\boldsymbol{\mu}}_k, \hat{\boldsymbol{\Sigma}}_k) \| \mathcal{N}(\boldsymbol{\mu}_k, \boldsymbol{\Sigma}))) \tag{36}$$

$$-\frac{\partial}{\partial \boldsymbol{\mu}_k}(\sum_{j \in \mathcal{B}_k} \hat{y}_j^\top \log(\frac{1}{\sqrt{(2\pi)^d |\boldsymbol{\Sigma}|}} \exp(-\frac{1}{2}(\mathbf{x}_j - \boldsymbol{\mu}_k)^\top \boldsymbol{\Sigma}^{-1}(\mathbf{x}_j - \boldsymbol{\mu}_k)))) \tag{37}$$

$$= -z_{i,k} \boldsymbol{\Sigma}^{-1}(\mathbf{x}_i - \boldsymbol{\mu}_k) + \frac{\partial}{\partial \boldsymbol{\mu}_k}(\beta \text{KL}(\mathcal{N}(\hat{\boldsymbol{\mu}}_k, \hat{\boldsymbol{\Sigma}}_k) \| \mathcal{N}(\boldsymbol{\mu}_k, \boldsymbol{\Sigma}))) \tag{38}$$

$$-\sum_{j \in \mathcal{B}_k} \hat{y}_{j,k} \boldsymbol{\Sigma}^{-1}(\mathbf{x}_j - \boldsymbol{\mu}_k) \tag{39}$$

$$\overset{(c)}{=} -z_{i,k} \boldsymbol{\Sigma}^{-1}(\mathbf{x}_i - \boldsymbol{\mu}_k) - \sum_{j \in \mathcal{B}_k} \hat{y}_{j,k} \boldsymbol{\Sigma}^{-1}(\mathbf{x}_j - \boldsymbol{\mu}_k) \tag{40}$$

$$+\frac{\partial}{\partial \boldsymbol{\mu}_k} \beta \frac{1}{2}(\log \frac{|\boldsymbol{\Sigma}|}{|\hat{\Sigma}_k|} - d + (\hat{\mu}_k - \mu_k)^\top \Sigma^{-1}(\hat{\mu}_k - \mu_k) + \text{Tr}(\Sigma^{-1}\hat{\Sigma}_k)) \tag{41}$$

$$= -z_{i,k} \boldsymbol{\Sigma}^{-1}(\mathbf{x}_i - \boldsymbol{\mu}_k) - \sum_{j \in \mathcal{B}_k} \hat{y}_{j,k} \boldsymbol{\Sigma}^{-1}(\mathbf{x}_j - \boldsymbol{\mu}_k) - \beta \boldsymbol{\Sigma}^{-1}(\hat{\boldsymbol{\mu}}_k - \boldsymbol{\mu}_k). \tag{42}$$

We obtain $\overset{(c)}{=}$ from Eq. (27). And, setting the partial derivative to zero, we can get:

$$\boldsymbol{\mu}_k = \frac{z_{i,k}\mathbf{x}_i + \sum_{j \in \mathcal{B}_k} \hat{y}_{j,k}\mathbf{x}_j + \beta\hat{\boldsymbol{\mu}}_k}{z_{i,k} + \sum_{j \in \mathcal{B}_k} \hat{y}_{j,k} + \beta} \tag{43}$$

$$\overset{(d)}{=} \frac{\sum_{j \in \mathcal{B}_k} \hat{y}_{j,k}\mathbf{x}_j + \beta\hat{\boldsymbol{\mu}}_k}{\sum_{j \in \mathcal{B}_k} \hat{y}_{j,k} + \beta} \tag{44}$$

$$= \boldsymbol{\mu}_k^*. \tag{45}$$

To enable efficient one-pass, closed-form online inference without iterative updates, we estimate the class mean by excluding the current instance $\mathbf{x}_i$ in $\overset{(d)}{=}$, helping prevent early overfitting and eliminating sub-iteration. The final mean vector is then expressed as a weighted combination with a scalar $\alpha \in [0,1]$:

$$\boldsymbol{\mu}_k^* \leftarrow \alpha\boldsymbol{\mu}_k' + (1-\alpha)\hat{\boldsymbol{\mu}}_k, \quad \text{where } \boldsymbol{\mu}_k' = \frac{\sum_{j \in \mathcal{B}_k} \hat{y}_{j,k}\mathbf{x}_j}{\sum_{j \in \mathcal{B}_k} \hat{y}_{j,k}}, \alpha = \frac{\sum_{j \in \mathcal{B}_k} \hat{y}_{j,k}}{\sum_{j \in \mathcal{B}_k} \hat{y}_{j,k} + \beta}. \tag{46}$$

### A.2.2 Proof of Our Closed-form Solution

We propose a closed-form label estimator (Eq. (8) in our main paper) for streaming scenarios by minimizing a regularized objective in Eq. (21). To derive a one-pass solution suitable for online inference, we compute the partial derivative of the objective with respect to the soft label assignment $z_i$. Since the Laplacian regularization term in $\mathcal{R}(z_i; \hat{y}_i)$ is concave, we incorporate the simplex

constraint $z_i \in \Delta^K$ to ensure that the solution lies within the probability simplex.

$$\frac{\partial \mathcal{L}_{\text{online}}}{\partial z_{i,k}} = \frac{\partial}{\partial z_{i,k}}(-z_i^\top \log \mathbb{P}_i) + \frac{\partial}{\partial z_{i,k}} \text{KL}(z_i \| \hat{y}_i) \tag{47}$$

$$- \frac{\partial}{\partial z_{i,k}} \sum_{j \in \mathcal{B}} w_{ij} z_i^\top \hat{y}_j + \frac{\partial}{\partial z_{i,k}} \lambda_i (\mathbb{I}_K^\top z_i - 1) \tag{48}$$

$$= -\log \mathbb{P}_{i,k} + \log z_{i,k} - \log \hat{y}_{i,k} + 1 - \sum_{j \in \mathcal{B}_k} w_{ij} \hat{y}_{j,k} + \lambda_i \tag{49}$$

$$= -\tilde{y}_{i,k} + \log z_{i,k} - \log \hat{y}_{i,k} + 1 - \sum_{j \in \mathcal{B}_k} w_{ij} \hat{y}_{j,k} + \lambda_i. \tag{50}$$

The first term in Eq. (49) is replaced by $-\tilde{y}_{i,k}$ because we can obtain $\tilde{y}_{i,k} \propto \log \mathbb{P}_{i,k}$ from Section A.1. And, setting the partial derivative to zero, we can get:

$$z_{i,k} = \hat{y}_{i,k} \cdot \exp\left(\tilde{y}_{i,k} + \sum_{j \in \mathcal{B}_k} w_{ij} \hat{y}_{j,k} - (1 + \lambda_i)\right), \tag{51}$$

$$\text{s.t.} \quad \sum_{k=1}^{K} z_{i,k} = 1. \tag{52}$$

We can get $\exp(-(1 + \lambda_i)) = 1/(\sum_{k=1}^{K} \hat{y}_{i,k} \cdot \exp(\tilde{y}_{i,k} + \sum_{j \in \mathcal{B}_k} w_{ij} \hat{y}_{j,k}))$. Bring it into Eq. (51), we can obtain the final optimal solution:

$$z_{i,k}^* = \frac{\hat{y}_{i,k} \cdot \exp\left(\tilde{y}_{i,k} + \sum_{j \in \mathcal{B}_k} w_{ij} \hat{y}_{j,k}\right)}{\sum_{k=1}^{K} \hat{y}_{i,k} \cdot \exp\left(\tilde{y}_{i,k} + \sum_{j \in \mathcal{B}_k} w_{ij} \hat{y}_{j,k}\right)}. \tag{53}$$

### A.3 Transductive Test-time Distribution Estimation

In the transductive setting, the entire test set $\mathcal{D}_u = \{\mathbf{x}_i\}_{i=1}^N$ is available during inference, allowing us to jointly optimize label distributions over all test instances rather than updating them sequentially. We extend the online regularized objective in Eq. (21) to a transductive objective as follows:

$$\mathcal{L}_{\text{trans}}(z, \boldsymbol{\mu}, \boldsymbol{\Sigma}) = -\sum_{i=1}^{N} z_i^\top \log \mathbb{P}_i + \sum_{i=1}^{N} \mathcal{R}(z_i; \hat{y}_i) + \sum_{i=1}^{N} \mathcal{R}(z_i; \mathcal{B}) \tag{54}$$

$$= -\sum_{i=1}^{N} z_i^\top \log \mathbb{P}_i + \sum_{i=1}^{N} \text{KL}(z_i \| \hat{y}_i) + \beta \sum_{k=1}^{K} \text{KL}\left(\mathcal{N}(\hat{\boldsymbol{\mu}}_k, \hat{\boldsymbol{\Sigma}}_k) \| \mathcal{N}(\boldsymbol{\mu}_k, \boldsymbol{\Sigma})\right) \tag{55}$$

$$- \sum_{j \in \mathcal{B}} \hat{y}_j^\top \log \mathbb{P}_j - \sum_{i=1}^{N} \sum_{j \in \mathcal{B}} w_{ij} z_i^\top \hat{y}_j. \tag{56}$$

Similar to the online case, we obtain the closed-form label estimator by taking the derivative of $\mathcal{L}_{\text{trans}}$ w.r.t. $z_{i,k}$:

$$\frac{\partial \mathcal{L}_{\text{trans}}}{\partial z_{i,k}} = \frac{\partial}{\partial z_{i,k}}\left(-\sum_{i=1}^{N} z_i^\top \log \mathbb{P}_i + \sum_{i=1}^{N} \mathcal{R}(z_i; \hat{y}_i) + \sum_{i=1}^{N} \mathcal{R}(z_i; \mathcal{B})\right) \tag{57}$$

$$= \frac{\partial}{\partial z_{i,k}}\left(-z_i^\top \log \mathbb{P}_i + \mathcal{R}(z_i; \hat{y}_i) + \mathcal{R}(z_i; \mathcal{B})\right) = \frac{\partial \mathcal{L}_{\text{online}}}{\partial z_{i,k}}. \tag{58}$$

With the results of Section A.2.2, we have:

$$z_{i,k}^* = \frac{\hat{y}_{i,k} \cdot \exp\left(\tilde{y}_{i,k} + \sum_{j \in \mathcal{B}_k} w_{ij} \hat{y}_{j,k}\right)}{\sum_{k=1}^{K} \hat{y}_{i,k} \cdot \exp\left(\tilde{y}_{i,k} + \sum_{j \in \mathcal{B}_k} w_{ij} \hat{y}_{j,k}\right)}. \tag{59}$$

Then, $\boldsymbol{\mu}_k$ is estimated by taking the derivative of $\mathcal{L}_{\text{trans}}$ w.r.t. $\boldsymbol{\mu}_k$:

$$\frac{\partial \mathcal{L}_{\text{trans}}}{\partial \boldsymbol{\mu}_k} = -\frac{\partial}{\partial \boldsymbol{\mu}_k}(\sum_{i=1}^N z_i^\top \log \mathbb{P}_i) + \frac{\partial}{\partial \boldsymbol{\mu}_k}(\beta \sum_{k=1}^K \text{KL}(\mathcal{N}(\hat{\boldsymbol{\mu}}_k, \hat{\boldsymbol{\Sigma}}_k)\|\mathcal{N}(\boldsymbol{\mu}_k, \boldsymbol{\Sigma}))) \tag{60}$$

$$-\frac{\partial}{\partial \boldsymbol{\mu}_k}(\sum_{j \in \mathcal{B}} \hat{y}_j^\top \log \mathbb{P}_j) \tag{61}$$

$$= -\sum_{i=1}^N \frac{\partial}{\partial \boldsymbol{\mu}_k}(z_i^\top \log \mathbb{P}_i) + \frac{\partial}{\partial \boldsymbol{\mu}_k}(\beta \sum_{k=1}^K \text{KL}(\mathcal{N}(\hat{\boldsymbol{\mu}}_k, \hat{\boldsymbol{\Sigma}}_k)\|\mathcal{N}(\boldsymbol{\mu}_k, \boldsymbol{\Sigma}))) \tag{62}$$

$$-\frac{\partial}{\partial \boldsymbol{\mu}_k}(\sum_{j \in \mathcal{B}} \hat{y}_j^\top \log \mathbb{P}_j) \tag{63}$$

$$= -\sum_{i=1}^N z_{i,k} \boldsymbol{\Sigma}^{-1}(\mathbf{x}_i - \boldsymbol{\mu}_k) - \beta \boldsymbol{\Sigma}^{-1}(\hat{\boldsymbol{\mu}}_k - \boldsymbol{\mu}_k) - \sum_{j \in \mathcal{B}_k} \hat{y}_{j,k} \boldsymbol{\Sigma}^{-1}(\mathbf{x}_j - \boldsymbol{\mu}_k). \tag{64}$$

Setting the Eq. (64) to zero, we have:

$$\boldsymbol{\mu}_k = \frac{\sum_{i=1}^N z_{i,k} \mathbf{x}_i + \sum_{j \in \mathcal{B}_k} \hat{y}_{j,k} \mathbf{x}_j + \beta \hat{\boldsymbol{\mu}}_k}{\sum_{i=1}^N z_{i,k} + \sum_{j \in \mathcal{B}_k} \hat{y}_{j,k} + \beta)} \tag{65}$$

$$\overset{(e)}{=} \frac{\sum_{i=1}^N \hat{y}_i \mathbf{x}_i + \sum_{j \in \mathcal{B}_k} \hat{y}_{j,k} \mathbf{x}_j + \beta \hat{\boldsymbol{\mu}}_k}{\sum_{i=1}^N \hat{y}_i + \sum_{j \in \mathcal{B}_k} \hat{y}_{j,k} + \beta} = \boldsymbol{\mu}_k^*. \tag{66}$$

In practice, to further improve efficiency and avoid the iterative updates required by MM-like algorithms, we obtain $\overset{(e)}{=}$ by substituting $z_i$ with the CLIP zero-shot predictions $\hat{y}_i$, yielding a one-pass estimate for class means:

$$\boldsymbol{\mu}_k^* \leftarrow \alpha \boldsymbol{\mu}_k' + (1-\alpha)\hat{\boldsymbol{\mu}}_k, \boldsymbol{\mu}_k' = \frac{\sum_{i=1}^N \hat{y}_{i,k} \mathbf{x}_i + \sum_{j \in \mathcal{B}_k} \hat{y}_{j,k} \mathbf{x}_j}{\sum_{i=1}^N \hat{y}_{i,k} + \sum_{j \in \mathcal{B}_k} \hat{y}_{j,k}}, \alpha = \frac{\sum_{i=1}^N \hat{y}_{i,k} + \sum_{j \in \mathcal{B}_k} \hat{y}_{j,k}}{\sum_{i=1}^N \hat{y}_{i,k} + \sum_{j \in \mathcal{B}_k} \hat{y}_{j,k} + \beta}. \tag{67}$$

### A.4   Limitations and Discussion

**Limitation.**   Our method assumes that class-conditional features follow Gaussian distributions with a shared covariance matrix. While this assumption simplifies the underlying data distribution and may not fully capture complex, multi-modal, or highly skewed patterns in real-world scenarios, it enables a key practical benefit. Specifically, the Gaussian assumption allows us to derive closed-form, backpropagation-free updates for both online and transductive test-time adaptation. This property is important for deploying vision-language models in environments that require low latency or have limited computational resources. Examples include mobile devices, robotic systems, and real-time inference settings, where gradient-based optimization is often infeasible.

**Discussion.**   To verify the core assumptions of our method that class-conditional features follow Gaussian distributions with a shared covariance matrix, we conduct two empirical analyses.

First, to examine the Gaussianity of class-conditional features, we adopt a projection-based testing strategy inspired by prior statistical literatures [35, 5, 6]. Classical multivariate normality tests [18, 43, 41] are known to break down when the feature dimension is large relative to the sample size (*e.g.*, $512/50 \gg 0$ in our case using CLIP-ViT/B-16 on ImageNet). To address this, we project high-dimensional data into low-dimensional subspaces by randomly selecting a small number of feature dimensions

Table 10: Projection-based normality test results across class-conditional features.

| | Low-dim | Freq of p>0.05 (%) ↑ | p-value Avg. ↑ |
|---|---|---|---|
| Henze–Zirkler | 2 | 100 | 0.39 |
| | 4 | 99.90 | 0.32 |
| | 6 | 99.00 | 0.27 |
| | 8 | 96.30 | 0.22 |
| | 10 | 92.90 | 0.19 |
| Shapiro–Wilk | 2 | 100 | 0.31 |
| | 4 | 100 | 0.21 |
| | 6 | 99.50 | 0.16 |
| | 8 | 96.30 | 0.13 |
| | 10 | 92.20 | 0.11 |

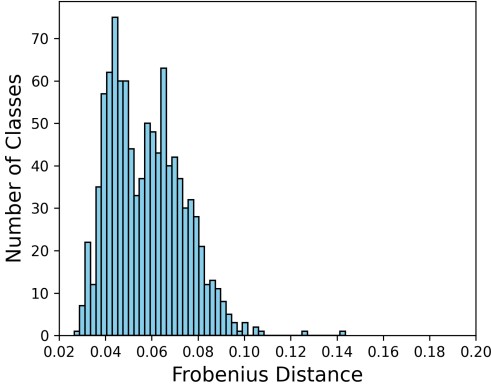 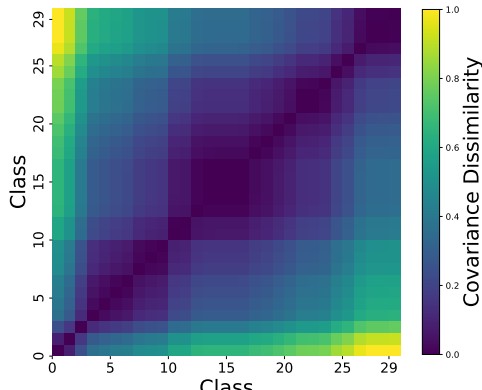

Figure 4: Comparison of covariance properties on ImageNet: (Left) shows Frobenius distance between class-wise $\Sigma_k$ and shared $\Sigma$, and (Right) shows class-wise covariance dissimilarity.

per class, and repeat this procedure 100 times. For each projected subset, we apply the Henze–Zirkler test [18] and Shapiro–Wilk test [41] for normality and report the average p-values across repetitions. As shown in Table 10, the average p-values frequently exceed 0.05, suggesting that class-conditional CLIP features are approximately Gaussian in projected subspaces. Furthermore, even when the data deviates from strict Gaussianity, our method remains effective as evidenced in Table 13, where it consistently achieves state-of-the-art performance. Existing works [27, 44, 48] also support the robustness of Gaussian discriminant analysis (GDA) against slight normality violations.

Second, we assess the validity of the shared covariance assumption. For each class, we compute its empirical covariance matrix and compare it to the pooled covariance using the Frobenius norm. As shown in Figure 4 (left), over 99.3% of the classes exhibit a Frobenius distance smaller than 0.1 from the pooled covariance, indicating strong alignment. Additionally, we visualize covariance matrices from several randomly selected classes in Figure 4 (right), and further quantify dissimilarity by comparing the trace values of class-wise covariance matrices. These heatmaps reveal consistent structural patterns across classes, providing qualitative evidence for the shared covariance structure. In summary, these findings collectively provide strong empirical support for the Gaussianity and shared covariance assumptions underlying our method.

To further compare with class-wise separate covariance matrices, we conducted additional experiments on ImageNet by replacing our shared covariance with class-specific covariance matrices, keeping all other settings identical. The results, summarized in Table 11, show that this leads to a significant drop in accuracy, alongside a sharp increase in computation time

Table 11: Evaluation with class-wise separate covariance matrices on ImageNet.

| Method | Time ↓ | Acc (%) ↑ | Gain (%) ↑ | Mem.(GB) ↓ |
|---|---|---|---|---|
| CLIP | 8m | 66.74 | - | 0.79 |
| ADAPT (Online) | 1h 11m | 70.91 | +4.17 | 0.93 |
| w/ Separate Cov. | 1h 44m | 53.80 | -12.94 | 2.89 |
| ADAPT (Trans.) | 0.73m | 71.56 | +4.82 | 3.37 |
| w/ Separate Cov. | 1.40m | 66.16 | -0.58 | 4.62 |

and memory usage. We attribute this performance degradation to data sparsity. Estimating a full covariance matrix for each class becomes unreliable with few high-confidence samples, leading to poor generalization and overfitting. In contrast, the shared covariance pools information across all classes, resulting in a much more stable and robust estimate, which is critical in online or data-scarce settings. Ultimately, this ablation confirms that our shared covariance assumption is not a limitation, but a crucial design choice. It strikes an effective trade-off between accuracy, robustness, and computational efficiency, making our method practical for real-world, resource-constrained scenarios.

**Remark.** To balance efficiency and stability in streaming scenarios with evolving data distributions, we exclude the current test sample $\mathbf{x}_i$ from online class mean updates. This decision is based on three key observations: (i) high-confidence samples are stored in the knowledge bank and will be incorporated later; (ii) low-confidence predictions contribute little and add noise; and (iii) moderately confident predictions are prone to errors, distorting class statistics. By omitting $\mathbf{x}_i$, we prevent early-stage prediction errors from propagating into the model. As shown in Table 8 (see main paper),

this approach offers significant computational savings with minimal accuracy loss, especially in the early stages of high uncertainty. In the transductive setting, we estimate class means $\boldsymbol{\mu}_k$ using the entire test set and accumulated high-confidence samples. Instead of relying on latent assignments $z_i$ optimized through iterative procedures (as in MM-based methods), we replace them with CLIP's zero-shot predictions $\hat{y}_i$. This design is motivated by three factors: (i) it avoids costly sub-iterations, enabling a one-pass, closed-form estimation of $\boldsymbol{\mu}_k$; (ii) with all test samples available in advance, computing $\hat{y}_i$ for the entire set is efficient and supports globally consistent estimation; (iii) although noisy, CLIP's soft predictions provide a reasonable proxy for class membership when aggregated. As shown in our ablations (Table 8 in the main paper), this heuristic achieves comparable or better performance than MM-based optimization, while being more stable and computationally efficient.

**Comparison with Other Distribution Estimation Methods.** We compare existing methods for estimating test-time distributions and demonstrate their limitations or inapplicability under the realistic online setting.

- GDA-CLIP [51]: This method estimates the class-wise distribution by computing the empirical mean and inverse covariance from an external training set $\mathcal{D}_s = \{\mathbf{x}_j\}_{j=1}^{S}$. The class mean is calculated as:

$$\boldsymbol{\mu}_k = \frac{\sum_{j \in \mathcal{D}_s} \mathbb{I}_{(y_j=k)} \mathbf{x}_j}{\sum_{j \in \mathcal{D}_s} \mathbb{I}_{(y_j=k)}}. \tag{68}$$

  Although the covariance is estimated using an empirical Bayes ridge-type estimator, this approach performs well only when the test distribution closely resembles the source data in its statistical properties. Crucially, it inherently assumes access to a labeled and stationary source distribution, which is unavailable in the TTA setting. As a result, it fails to adapt under distribution shifts, and the estimated statistics can quickly become outdated or misaligned with the continuously evolving test data stream.

- Frolic [69]: This method directly adopts CLIP's class prototypes as class-wise means, *i.e.*, $\boldsymbol{\mu}_k = \mathbf{t}_k$, and estimates the shared covariance $\boldsymbol{\Sigma}$ from the marginal distribution via the expectation and second-order moment, as follows:

$$\boldsymbol{\Sigma} = \frac{1}{N} \sum_{i \in \mathcal{D}_u} \mathbf{x}_i \mathbf{x}_i^\top - \frac{1}{K} \sum_{k=1}^{K} \boldsymbol{\mu}_k \boldsymbol{\mu}_k^\top. \tag{69}$$

  However, this method requires complete access to the entire target dataset $\mathcal{D}_u = \{\mathbf{x}_i\}_{i=1}^{N}$, making it unsuitable for online or streaming scenarios where test samples arrive sequentially and future data is inaccessible.

- TransCLIP [59]: This method estimates the parameters $\boldsymbol{\mu}_k$ and $\boldsymbol{\Sigma}$ via a Majorize-Minimize (MM) optimization procedure, which iteratively refines the predicted soft-assignments $z$ and the distribution parameters:

$$\boldsymbol{\mu}_k = \frac{\sum_{i \in \mathcal{D}_u} z_{i,k} \mathbf{x}_i}{\sum_{i \in \mathcal{D}_u} z_{i,k}}, \tag{70}$$

$$\mathrm{diag}(\boldsymbol{\Sigma}) = \frac{\sum_{i \in \mathcal{D}_u} z_{i,k} (\mathbf{x}_i - \boldsymbol{\mu}_k)^2}{N}. \tag{71}$$

  It simultaneously relies on full access to the entire target dataset $\mathcal{D}_u$ and involves computationally expensive inner-loop MM iterations to achieve convergence. These two limitations significantly hinder its practicality in real-time or resource-constrained online scenarios, where access to future samples is restricted and fast adaptation is essential.

- ADAPT (Ours): In contrast to the above methods, ADAPT progressively estimates class-wise means and a shared covariance matrix without requiring supervision or access to source data (see Section A.2 and Section A.3). This leads to a closed-form, training-free solution that is naturally compatible with both online and transductive test-time adaptation settings. Moreover, ADAPT incurs minimal computational overhead and seamlessly adapts to distributional shifts in streaming scenarios.

Table 12: Top-1 accuracy (%) comparison on natural distribution shift task with CLIP-RN50 backbone under both online and transductive protocols.

| | Method | BP-free | ImageNet | ImageNet-A | ImageNet-V | ImageNet-R | ImageNet-S | OOD Avg. | Avg. |
|---|---|---|---|---|---|---|---|---|---|
| Online | CLIP [39] | - | 58.16 | 21.83 | 51.41 | 56.15 | 33.37 | 40.69 | 44.18 |
| | TPT [33] | ✗ | 60.74 | 26.67 | 54.70 | 59.11 | 35.09 | 43.89 | 47.26 |
| | DiffTPT [9] | ✗ | 60.80 | 31.06 | 55.80 | 58.80 | 37.10 | 45.69 | 48.71 |
| | C-TPT [55] | ✗ | 60.20 | 23.40 | 54.70 | 58.00 | 35.10 | 42.80 | 46.28 |
| | DMN [65] | ✗ | **63.87** | 28.57 | 56.12 | 61.44 | 39.84 | 46.49 | 49.97 |
| | TPS [46] | ✗ | 62.27 | 29.80 | **60.04** | 55.49 | 35.74 | 45.27 | 48.67 |
| | DPE [61] | ✗ | 63.41 | 30.15 | 56.72 | **63.72** | 40.03 | 47.66 | 50.81 |
| | DynaPrompt [54] | ✗ | 61.56 | 27.84 | 55.12 | 60.63 | 35.64 | 44.81 | 48.16 |
| | BCA [67] | ✓ | 61.81 | 30.35 | 56.58 | 62.89 | 38.04 | 46.97 | 49.93 |
| | TDA [21] | ✓ | 61.35 | 30.29 | 55.54 | 62.58 | 38.12 | 46.63 | 49.58 |
| | Dota [12] | ✓ | 61.82 | 30.81 | 55.27 | 62.81 | 37.52 | 46.60 | 49.65 |
| | ADAPT | ✓ | 62.16 | **33.08** | 55.97 | 62.69 | **40.21** | **47.99** | **50.82** |
| Trans. | TransCLIP [59] | ✓ | 58.00 | 21.93 | 51.54 | 35.15 | **52.79** | 40.35 | 43.88 |
| | ADAPT | ✓ | **62.94** | **33.72** | **56.57** | **63.11** | 41.19 | **48.65** | **51.51** |

Table 13: Top-1 accuracy (%) comparison on corruption robustness task with CLIP-RN50 backbone under both online and transductive protocols.

| | Method | Blur | | | | Weather | | | | Digital | | | | Noise | | | Avg. |
|---|---|---|---|---|---|---|---|---|---|---|---|---|---|---|---|---|---|
| | | Defo. | Glas. | Moti. | Zoom | Snow | Fros. | Fog | Brig. | Cont. | Elas. | Pix. | JPEG | Gauss. | Shot | Impu. | |
| Online | CLIP [39] | 9.54 | 3.40 | 7.46 | 12.62 | 12.29 | 15.72 | 22.08 | 41.69 | 6.24 | 4.67 | 11.01 | 14.24 | 2.43 | 3.07 | 2.52 | 11.27 |
| | TPT [33] | 8.02 | 2.74 | 5.34 | 10.97 | 10.59 | 12.92 | 16.17 | 35.67 | 4.45 | 3.73 | 11.56 | **16.68** | 1.43 | 1.94 | 1.42 | 9.58 |
| | DiffTPT [9] | 10.50 | 3.90 | **8.62** | 12.74 | 11.31 | 15.03 | 19.64 | 37.73 | 4.98 | 4.74 | **13.58** | 16.56 | 2.69 | 3.59 | 2.08 | 11.18 |
| | TDA [21] | 9.84 | 4.4 | 7.38 | 13.74 | 13.74 | 17.16 | 23.76 | 44.16 | 7.00 | 5.79 | 11.24 | 15.26 | 2.54 | 3.26 | 2.72 | 12.13 |
| | ADAPT | **10.54** | 4.44 | 8.57 | 14.34 | 13.85 | 17.84 | 24.56 | 45.67 | 7.76 | 5.85 | 11.96 | 15.86 | 2.91 | 3.77 | 2.92 | 12.72 |
| Trans. | ZLaP [20] | 10.3 | 3.54 | 7.99 | 13.47 | 13.66 | 17.15 | 23.2 | 44.67 | 6.55 | 5.15 | 11.61 | 14.23 | 1.17 | 2.33 | 1.65 | 11.82 |
| | TransCLIP [59] | 9.38 | 4.17 | 7.14 | 12.42 | 11.87 | 15.04 | 23.93 | 44.33 | 6.45 | 6.11 | 11.03 | 14.85 | 2.87 | 3.14 | 3.13 | 11.72 |
| | ADAPT | **12.26** | **6.11** | **10.92** | **17.02** | **16.67** | **20.62** | **27.77** | **47.31** | **9.20** | **8.56** | **14.61** | **18.15** | **4.05** | **4.80** | **4.03** | **14.81** |

# B   More Experimental Results

## B.1   Experiments with CLIP-RN50 Backbone

**Task 1: Natural Distribution Shift.**   Table 12 reports the performance on natural distribution shift benchmarks using the CLIP-RN50 backbone under both online and transductive settings. In the online setting, ADAPT outperforms all prior backpropagation-free and optimization-based methods, achieving the highest average accuracy of 50.82%. It even surpasses strong transductive methods such as TransCLIP, despite not accessing the full target set. In the transductive setting, where all test samples are available, ADAPT further improves to 51.51%, exceeding TransCLIP by 7.46%. It also ranks first on OOD benchmarks, demonstrating consistent robustness under domain shifts.

**Task 2: Corruption Robustness.**   We further evaluate the performance on corruption robustness benchmarks, with results presented in Table 13. Using the CLIP-RN50 backbone, ADAPT achieves the highest average accuracy of 12.72% in the online setting, outperforming all prior backpropagation-free and optimization-based methods. In the transductive setting, ADAPT further improves to 14.81% and surpasses all compared methods. It consistently ranks first across all 15 corruption types, demonstrating strong resilience to diverse perturbations and confirming its robustness under severe domain shifts.

**Task 3: Fine-Grained Categorization.**   Finally, we evaluate the ADAPT 's adaptation on fine-grained categorization benchmarks, as shown in Table 14. In the Online setting, ADAPT achieves 62.82% average accuracy, which is competitive with the strongest methods like DMN, while maintaining the advantage of being backpropagation-free. In the transductive setting, ADAPT achieves the highest accuracy of 64.64%, outperforming prior state-of-the-art methods such as TransCLIP and StatA. It ranks at the top in most datasets, demonstrating strong generalization to subtle inter-class differences. These results highlight the effectiveness and robustness of our framework under fine-grained and distribution-shifted conditions.

Table 14: Top-1 accuracy (%) comparison on fine-grained categorization task with CLIP-RN50 backbone under both online and transductive protocols.

| | Method | BP-free | Aircraft | Caltech | Cars | DTD | EuroSAT | Flower | Food101 | Pets | Sun397 | UCF101 | Avg. |
|---|---|---|---|---|---|---|---|---|---|---|---|---|---|
| **Online** | CLIP [39] | - | 15.66 | 85.88 | 55.70 | 40.37 | 23.69 | 61.75 | 73.97 | 83.57 | 58.80 | 58.84 | 55.82 |
| | TPT [33] | ✗ | 17.58 | 87.02 | 58.46 | 40.84 | 28.33 | 62.69 | 74.88 | 84.49 | 61.46 | 60.82 | 57.66 |
| | DiffTPT [9] | ✗ | 17.60 | 86.89 | **60.71** | 40.72 | 41.04 | 63.53 | **79.21** | 83.40 | 62.72 | 62.67 | 59.85 |
| | C-TPT [55] | ✗ | 17.00 | 86.90 | 56.50 | 42.20 | 27.80 | 65.20 | 74.70 | 84.10 | 61.00 | 59.70 | 57.51 |
| | DMN [65] | ✗ | **22.77** | 90.14 | 60.02 | 50.41 | 48.72 | 67.93 | 76.70 | 86.78 | 64.39 | **65.34** | **63.32** |
| | TPS [29] | ✗ | 18.30 | 89.80 | 59.40 | 48.40 | 24.30 | 68.20 | 76.20 | 84.40 | 62.70 | 64.30 | 59.60 |
| | DPE [61] | ✗ | 19.80 | **90.83** | 59.26 | 50.18 | 41.67 | 67.60 | 77.83 | 85.97 | 64.23 | 61.98 | 61.94 |
| | BCA [67] | ✓ | 19.89 | 89.70 | 58.13 | 48.58 | 42.12 | 66.30 | 77.19 | 85.58 | 63.38 | 63.51 | 61.44 |
| | TDA [21] | ✓ | 17.61 | 89.70 | 57.78 | 43.74 | 42.11 | 68.74 | 77.75 | 86.18 | 62.53 | 64.18 | 61.03 |
| | Dota [7] | ✓ | 18.06 | 88.84 | 58.72 | 45.80 | 47.15 | 68.53 | 78.61 | **87.33** | 63.89 | 65.08 | 62.20 |
| | ADAPT | ✓ | 18.00 | 89.37 | 58.38 | **51.89** | 50.47 | 70.04 | 75.57 | 86.43 | **64.94** | 63.12 | 62.82 |
| **Trans.** | TransCLIP [59] | ✓ | 16.60 | 88.60 | 57.90 | 47.80 | **59.60** | 72.20 | **78.00** | 89.30 | 64.20 | **68.80** | 64.30 |
| | StatA [58] | ✓ | 16.00 | 87.30 | 58.20 | 48.50 | 50.50 | 67.70 | 77.90 | 87.70 | 64.30 | 67.50 | 62.56 |
| | ADAPT | ✓ | **19.53** | **90.99** | **61.46** | **55.73** | 46.26 | **74.14** | 76.97 | 87.95 | **66.66** | 66.75 | **64.64** |

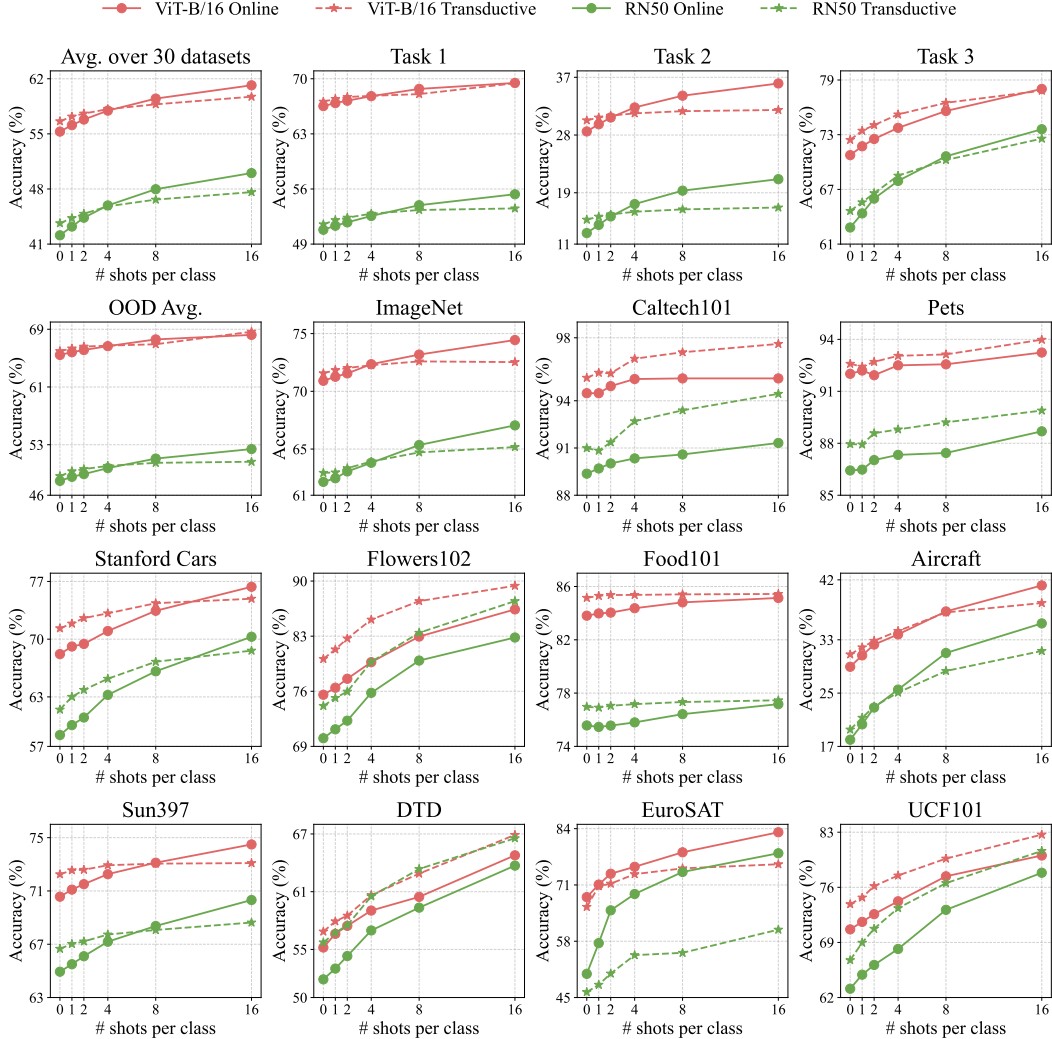

Figure 5: Results of few-shot classification across 30 datasets. We evaluate our method under both online and transductive protocols with 0, 1, 2, 4, 8, and 16-shot settings.

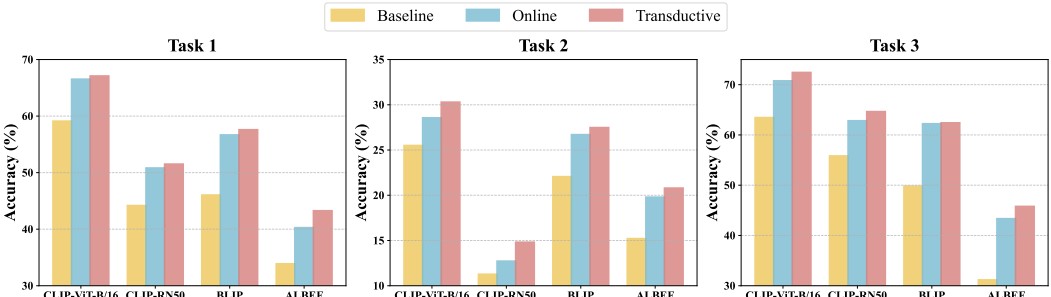

Figure 6: Performance comparison of proposed ADAPT on different VLMs.

## B.2 Few-shot Adaptation

Figure 5 presents few-shot classification results of our ADAPT across three tasks covering 30 datasets. We evaluate performance under both online and transductive adaptation protocols, using 0, 1, 2, 4, 8, and 16 shots per class. To assess architectural generality, we conduct experiments with two backbone variants: CLIP-ViT-B/16 and CLIP-RN50.

Overall, results show a consistent performance increase with more labeled samples, confirming the scalability and effectiveness of our method. Results with CLIP-ViT-B/16 consistently outperform RN50 across all shot counts, particularly in low-shot regimes, suggesting stronger feature representations. Additionally, transductive adaptation variants exhibit consistently better performance than their online counterparts. This improvement can be attributed to the transductive setting's ability to leverage the entire test set as a global context, thereby facilitating more informed label predictions and reducing uncertainty in classification. Notably, Task 3 requires finer-grained discrimination due to subtle intra-class variations, making accurate classification particularly dependent on detailed visual cues. In this context, the few-shot setting offers class-specific supervision that enhances the model's ability to capture these nuances. As a result, our method exhibits substantial performance gains on Task 3, with consistent improvements observed across 10 representative datasets. These findings underscore our method's capacity to adapt to complex recognition challenges and its effectiveness across varying adaptation regimes.

## B.3 Evaluation with Different VLMs

We conduct a comprehensive evaluation of our ADAPT across a diverse set of vision-language models (VLMs), including CLIP (ViT-B/16 and RN50) [39], BLIP [25], and ALBEF [26]. Figure 6 presents the performance comparison across three representative tasks: Task 1 assessing natural distribution shifts, Task 2 evaluating robustness to synthetic corruptions, and Task 3 focusing on fine-grained categorization challenges.

Our results demonstrate that ADAPT consistently improves performance over baseline methods across all tasks and model architectures. The online adaptation variant already delivers notable gains by leveraging sequential test data, while the transductive variant further enhances results by exploiting global information from the entire test set. Importantly, these improvements hold true not only for stronger VLMs like CLIP-ViT-B/16 but also for comparatively lighter models such as ALBEF, indicating that ADAPT is broadly applicable and effective regardless of the underlying model capacity. This versatility highlights the practical value of our approach for enhancing test-time adaptation across diverse vision-language architectures and real-world scenarios.

## B.4 Additional Analysis

**Further Analysis on Ablation Study.** Table 15 presents a detailed ablation study of three key components in our ADAPT: (i) the knowledge bank $\mathcal{B}$, (ii) the adaptive class-wise mean $\mu$, and (iii) shared covariance $\Sigma$. We compare eight configurations by selectively enabling or disabling these components and report performance across all three tasks. In this setup, disabling $\mu$ means using CLIP's original class prototypes as the fixed class means, while disabling $\Sigma$ corresponds to using a fixed identity matrix as the shared covariance.

The upper block of the table (Rows 1–4) corresponds to knowledge bank-free settings, where $\mathcal{B}$ is not used. The first row represents the baseline CLIP model without any adaptation. Updating only $\mu$ (Row 3) provides modest improvements (*e.g.*, +2.43% on Task 1), likely due to better-aligned class centers. However, updating only $\Sigma$ (Row 2) leads to substantial performance drops (*e.g.*, down to 9.58% on Task 2), indicating that estimating covariance from noisy test-time predictions alone is highly unstable and unreliable.

Table 15: Ablation study on components.

| $\mathcal{B}$ | Update $\mu$ | Update $\Sigma$ | Task 1 | Task 2 | Task 3 |
|---|---|---|---|---|---|
| ✗ | ✗ | ✗ | 59.11 | 25.50 | 63.45 |
| ✗ | ✗ | ✓ | 49.64 | 9.58 | 60.02 |
| ✗ | ✓ | ✗ | 61.54 | 25.42 | 67.03 |
| ✗ | ✓ | ✓ | 49.65 | 9.58 | 60.04 |
| ✓ | ✗ | ✗ | 64.89 | 25.08 | 67.06 |
| ✓ | ✗ | ✓ | 64.05 | 19.49 | 67.95 |
| ✓ | ✓ | ✗ | 65.27 | 25.67 | 67.43 |
| ✓ | ✓ | ✓ | **66.53** | **28.56** | **70.76** |

The lower block (Rows 5–8) introduces the knowledge bank $\mathcal{B}$. Even without updating $\mu$ or $\Sigma$ (Row 5), the model significantly outperforms all memory-free variants, validating the effectiveness of the regularization term $\mathcal{R}(z_i; \mathcal{B})$, which encourages alignment with high-confidence stored features. Incorporating adaptation for either $\mu$ or $\Sigma$ further improves results, showing that both distributional statistics offer complementary cues for more accurate estimation. The best performance is achieved by jointly adapting both $\mu$ and $\Sigma$, which fully leverages the knowledge bank for robust distribution modeling, with Task 2 reaching 28.56%. These results highlight the synergy between adaptive statistics and memory-guided estimation for stable TTA.

**Robustness under Significant Shifts.** To evaluate our method's robustness under significant distribution shifts, we benchmarked ADAPT against TDA [21], a state-of-the-art memory-based method. The evaluation was conducted using synthetic corruptions across five levels of increasing severity, with detailed results presented in Table 16. The results show that while all methods degrade as the shift intensifies,

Table 16: Robustness evaluation on ImageNet-C across five levels of synthetic corruption.

| Severity Level | 1 | 2 | 3 | 4 | 5 |
|---|---|---|---|---|---|
| CLIP | 59.68 | 52.97 | 46.79 | 37.51 | 25.50 |
| TDA | 60.42 | 54.15 | 48.35 | 36.84 | 28.34 |
| ADAPT (Online) | 61.37 | 54.70 | 48.61 | 39.28 | 28.56 |
| ADAPT (Trans.) | 62.04 | 55.51 | 49.68 | 40.76 | 30.29 |

ADAPT consistently outperforms TDA across all severity levels. This provides strong evidence that our proposed mechanism is more resilient to noisy pseudo-labels, validating its superior robustness for practical applications involving severe, real-world distribution shifts.

**Robustness under Low-confidence Scenarios.** To assess the impact of having a limited number of high-confidence samples per class (denoted as $N$), we analyze our method's performance in this constrained setting. The results, presented in Table 17, reveal distinct behaviors for our online and transductive approaches. Our online method, while sensitive to extremely low sample counts ($N = 2$), recovers quickly and achieves robust performance with as few as 4–6 samples per class. However, we acknowledge a potential efficiency challenge: if the test stream begins with many low-confidence predictions, it takes longer

Table 17: Evaluation with very few high-confidence samples. Where N means the number of high-confidence samples per class.

| | N | 2 | 4 | 6 | 8 | 10 |
|---|---|---|---|---|---|---|
| Online | Task 1 | 59.61 | 64.19 | 65.50 | 66.10 | 66.33 |
| | Task 2 | 22.95 | 26.00 | 27.26 | 27.93 | 28.22 |
| | Task 3 | 63.87 | 67.46 | 69.04 | 69.89 | 70.31 |
| Trans. | Task 1 | 67.02 | 67.11 | 67.09 | 67.10 | 67.04 |
| | Task 2 | 30.26 | 30.29 | 30.29 | 30.23 | 30.23 |
| | Task 3 | 72.15 | 72.21 | 72.43 | 72.24 | 72.25 |

to collect reliable samples for the memory bank, which may slightly delay adaptation and degrade performance. Notably, our transductive ADAPT demonstrates exceptional robustness, maintaining high and stable performance even with just two high-confidence samples per class ($N = 2$). We attribute this resilience to its ability to leverage the global structure of the entire test batch, which regularizes parameter estimates and ensures strong performance even under severe data scarcity.

