# OpenReview forum: "Backpropagation-Free Test-Time Adaptation via Probabilistic Gaussian Alignment"
_NeurIPS.cc/2025/Conference — NeurIPS 2025 poster_

### Official Review · Reviewer_o7Us · 2025-06-29

**Clarity:** 3
**Significance:** 2
**Originality:** 2
**Rating:** 4
**Confidence:** 4

**Summary:**

This paper presents ADAPT, a novel method for test-time adaptation (TTA) that is both backpropagation-free and distribution-aware. The core idea is to reframe TTA as a probabilistic inference problem, where class-conditional feature distributions are modeled as Gaussians. This allows for a closed-form, training-free solution that efficiently adapts vision-language models to new test distributions using dynamically updated "knowledge banks" of high-confidence samples.

**Questions:**

1. What is Oracle ADAPT and how does it differ from ADAPT in the transductive setting?
2. Consider adding memory and computation analysis to strengthen ADAPT's contribution. This should favor ADAPT since it avoids backpropagation.
3. Have you explored class-specific covariance matrices? What's the accuracy vs. computational complexity trade-off?
4. Why exclude the current test sample from class mean updates in online but include all samples (via soft labels) in transductive settings?
5. How does the knowledge bank perform on high inter-class similarity datasets? Could low confidence across all classes stall updates and hurt adaptation?
6. How did you tune the regularization parameter β in Eq. 5, and how sensitive is performance to its value?

**Ethical Concerns:**

["NO or VERY MINOR ethics concerns only"]

**Final Justification:**

The rebuttal and response addressed most of my concerns. I maintain my positive rating.

**Limitations:**

yes

**Quality:**

3

**Strengths And Weaknesses:**

# Strengths
I think the main strength of this work lies in its efficiency and practicality. By eliminating the need for backpropagation and iterative optimization, ADAPT presents a compelling solution for real-time or resource-constrained scenarios, a significant advantage over many existing TTA methods. I also found the framework's flexibility to be a major plus; its ability to operate in both online and transductive settings without requiring source data makes it broadly applicable.

# Weaknesses

Despite the strong results, I have a few concerns:

1. The method's core assumption is that class features follow a Gaussian distribution with a shared covariance matrix (lines 128-129) and a uniform class prior (line 134),  it may be too simplistic for real-world data distributions that could be multi-modal or highly skewed. As noted in prior work [1, 2, 3], the assumption of uniform prior is often violated in the real world, which could degrade the performance of methods requiring such an assumption. An analysis of potential failure cases would strengthen the paper.

2. The literature review contains several critical gaps that weaken the paper's novelty claims. The backpropagation-free TTA approach using pre-trained classifier weights as prototype anchors closely parallels [4]'s vision-only methodology—creating linear layers from CLIP's text embeddings is essentially a direct extension of their approach. Other backpropagation-free TTA methods [6, 7] for vision or vision-language model are missing. Additionally, the "Knowledge banks" mechanism resembles those in [3, 5] ([5] is cited but not discussed), and ADAPT's update strategy (e.g., Eq. 9) appears very similar to existing work. The paper requires proper citations and clearer differentiation from these prior works to establish its contributions.

[1] NOTE: Robust Continual Test-time Adaptation Against Temporal Correlation (NeurIPS 2022)

[2] Robust Test-Time Adaptation in Dynamic Scenarios (CVPR 2023)

[3] DPCore: Dynamic Prompt Coreset for Continual Test-Time Adaptation (ICML 2025)

[4] Test-Time Classifier Adjustment Module for Model-Agnostic Domain Generalization (NeurIPS 2021)

[5] DynaPrompt: Dynamic Test-Time Prompt Tuning (ICLR 2025)

[6] Test-Time Model Adaptation with Only Forward Passes (ICML 2024)

[7] Black-Box Test-Time Prompt Tuning for Vision-Language Models (AAAI 2025)

---

> ### Author Rebuttal · Authors · 2025-07-29
>
> We thank the reviewer for their thorough analysis and constructive feedback.
>
> ---
> > **W1: Discussion on the assumption of Gaussian distributions with shared covariance.**
>
> We acknowledge that assuming a unimodal Gaussian distribution with a shared covariance may not capture the full complexity of all hypothetical, multi-modal data structures. For instance, the performance on the ImageNet-C corruption benchmark is significantly lower compared to clean ImageNet. However, as we discuss in **Appendix A.4** (Limitations and Discussion), this assumption was a deliberate design choice made to prioritize practicality and efficiency, and it is supported by strong empirical evidence.
>
> Furthermore, as we show later in **Q3 of our rebuttal**, a more complex model does not inherently yield better results. We found that adopting class-specific separate covariance matrices actually degrades performance, while significantly increasing memory usage and inference time. This suggests that **our simpler assumption strikes a crucial and effective trade-off between model fidelity, robustness, and computational efficiency**. This balance is precisely what enables ADAPT to be, as the reviewer insightfully highlighted, "a compelling solution for real-time or resource-constrained scenarios, a significant advantage over many existing TTA methods."
>
> ---
> > **W2: More comparisons with prior works.**
>
> We extend the comparisons to include most of the works mentioned by the reviewer, excluding DPCore [1] and FOA [4], which require source data and thus violate ADAPT's source-free protocol. Instead, recent SOTA training-free methods such as AWT [6] and RA-TTA [7] are included. A more detailed comparison and discussion will be provided in the final version.
>
> | | Method | Venue | ImageNet | ImageNet-A | ImageNet-V | ImageNet-R | ImageNet-S | OOD Avg. | Avg. |
> | :--- | :--- | :--- | :--- | :--- | :--- | :--- | :--- | :--- | :--- |
> | | T3A [2] | NeurIPS 2021 | 66.09 | 47.54 | 61.34 | 69.24 | 46.82 | 56.24 | 58.21 |
> | | DynaPrompt [3] | ICLR 2025 | 69.61 | 56.17 | 64.67 | 78.17 | 48.22 | 61.81 | 63.37 |
> | | B²TPT [5] | AAAI 2025 | 69.57 | 55.26 | 65.40 | 78.64 | 49.53 | 62.21 | 63.68 |
> | | AWT [6] | NeurIPS 2024 | 71.32 | 60.33 | 65.15 | 80.64 | 51.60 | 64.43 | 65.81 |
> | | RA-TTA [7] | ICLR 2025 | 70.58 | 59.21 | 64.16 | 79.68 | 50.83 | 63.47 | 64.89 |
> | | ADAPT (Online) | | 70.91 | 63.32 | 64.64 | 80.66 | 53.13 | 65.44 | 66.53 |
> | | ADAPT (Trans.) | | 71.56 | 63.77 | 65.59 | 80.64 | 53.87 | 65.97 | 67.09 |
>
> **References**:
>
> [1] DPCore: Dynamic Prompt Coreset for Continual Test-Time Adaptation (ICML 2025)
>
> [2] Test-Time Classifier Adjustment Module for Model-Agnostic Domain Generalization (NeurIPS 2021)
>
> [3] DynaPrompt: Dynamic Test-Time Prompt Tuning (ICLR 2025)
>
> [4] Test-Time Model Adaptation with Only Forward Passes (ICML 2024)
>
> [5] Black-Box Test-Time Prompt Tuning for Vision-Language Models (AAAI 2025)
>
> [6] AWT: Transferring Vision-Language Models via Augmentation, Weighting, and Transportation (NeurIPS 2024)
>
> [7] RA-TTA: Retrieval-Augmented Test-Time Adaptation for Vision-Language Models (ICLR 2025)
>
> ---
> > **Q1: Clarification on the Oracle ADAPT.**
>
> As noted in **Section 4.3**, Oracle ADAPT uses the **same setup as transductive ADAPT, but with one key difference: it leverages test set ground-truth labels to estimate the target distribution**. While this is not feasible in real-world TTA scenarios, **this setting serves as an upper bound for performance**. In contrast, our standard ADAPT is fully unsupervised and label-free.
>
> As shown in **Table 4** of our main paper, our standard method achieves results within **5%** of this oracle, showing that ADAPT can closely approximate ideal adaptation performance without supervision. This highlights the strength and practicality of our approach in real-world test-time adaptation settings.
>
> ---
> > **Q2: Efficiency analysis.**
>
> We already provided efficiency analysis in **Table 6** of our main paper. Compared with optimization-based methods (e.g., TPT, DiffTPT, and TPS) and training-free methods like TDA, ADAPT achieves the largest performance gains on ImageNet with **just 1h 11m and 0.93GB in the online scenario**.
>
> Notably, our **transductive ADAPT** builds the knowledge bank once and predicts all samples at once, requiring only **0.73 minutes** and **3.37GB**. These results highlight ADAPT as a practical solution balancing accuracy and efficiency.
>
> ---
> > **Q3: Comparison with separate covariance matrices.**
>
> We conducted an additional experiment on ImageNet by **replacing our shared covariance matrix with class-specific covariance matrices**, while keeping all other settings identical.
>
> As shown in the table below, this variant leads to **a noticeable drop in accuracy, along with substantial increases in both computation time and memory usage**. We attribute this performance degradation to data sparsity. When only a few high-confidence samples are available for each class, estimating a full covariance matrix becomes unreliable. This can lead to overfitting and poor generalization.
>
> In contrast, using a shared covariance matrix allows the model to pool information across all classes, yielding a more stable and robust estimate. This design is especially beneficial in online or data-scarce scenarios, where efficiency and generalization are crucial.
>
> These results further validate our design choice. **Our shared covariance not only enhances performance but also supports ADAPT's efficiency, low latency, and applicability across diverse settings, making it a practical solution for real-world deployment**.
>
> | Method | Time | Acc | Gains | Mem. (GB) |
> | :--- | :--- | :--- | :--- | :--- |
> | CLIP | 8m | 66.74 | - | 0.79 |
> | ADAPT (Online) | 1h 11m | 70.91 | 4.17 | 0.93 |
> | w/ Separate Cov. | 1h 44m | 53.80 | -12.94 | 2.89 |
> | ADAPT (Trans.) | 0.73m | 71.56 | 4.82 | 3.37 |
> | w/ Separate Cov. | 1.40m | 66.16 | -0.58 | 4.62 |
>
> ---
> > **Q4: Why exclude the current test sample from class mean updates in the online but include all samples in the transductive settings?**
>
> We already analyzed and discussed this in the last paragraph of **Section 3.2 (Remark)** of our main paper, with further details provided in **Appendix A.4 (Remark)**. We summarize the key points as follows.
>
> In the online setting, we exclude the current test sample from class mean updates to **enable a one-pass, iteration-free solution**. High-confidence samples are automatically stored in the knowledge bank and used for future updates without additional processing. So, this design also avoids injecting noise from low- or mid-confidence predictions. As shown in **Table 8** of our main paper, this strategy **achieves comparable accuracy to the iterative version (with the current test sample) while achieving a 4× speedup on ImageNet in the online scenario**.
>
> In contrast, **the transductive setting assumes access to the entire test set beforehand, allowing all samples to contribute to a more accurate estimate of the class distribution**. To leverage global consistency without iterative optimization procedures such as Majorization-Minimization, we replace hard latent assignments with soft labels.
>
> As demonstrated in **Table 8** of our main paper, this choice leads to an effective balance between accuracy and efficiency.
>
> ---
> > **Q5: On the robustness to high inter-class similarity and low-confidence scenarios.**
>
> Thank you for raising this question regarding challenging cases with high inter-class similarity and limited confident samples.
>
> As demonstrated in the **fine-grained categorization results in Table 4** of the main paper, where distinguishing between classes is particularly difficult, ADAPT achieves state-of-the-art results, highlighting its effectiveness even when classes are highly similar.
>
> To further assess the impact of **limited high-confidence samples**, we vary the number of available samples per class (denoted as **N**) in this constrained setting. The results in the table below show that our method remains robust. Specifically, **Online ADAPT performs reliably with as few as 4–6 confident samples per class, and Transductive ADAPT maintains competitive accuracy even when provided with only two confident samples per class**.
>
> We acknowledge, however, that extremely low confidence across all classes presents a challenge, especially for the efficiency of our ADAPT in the online scenario. As shown in **Table 9** of our main paper, if the test stream begins with many low-confidence predictions, it takes longer to collect reliable samples for the memory bank, which may slightly delay adaptation and degrade performance.
>
> |   | N | 2 | 4 | 6 | 8 | 10 |
> | :--- | :--- | :--- | :--- | :--- | :--- | :--- |
> | Online | Task 1 | 59.61 | 64.19 | 65.50 | 66.10 | 66.33 |
> | | Task 2 | 22.95 | 26.00 | 27.26 | 27.93 | 28.22 |
> | | Task 3 | 63.87 | 67.46 | 69.04 | 69.89 | 70.31 |
> | Trans. | Task 1 | 67.02 | 67.11 | 67.09 | 67.10 | 67.04 |
> | | Task 2 | 30.26 | 30.29 | 30.29 | 30.23 | 30.23 |
> | | Task 3 | 72.15 | 72.21 | 72.43 | 72.24 | 72.25 |
>
> ---
> > **Q6: Analysis of parameter $\beta$.**
>
> **The parameter $\beta$ in Eq. 5 serves to map the scalar $\alpha$ into the range [0, 1]**, which is then applied in computing the class mean in Eq. 9 and Eq. 12. The value of $\alpha$ controls the balance between the CLIP prior and the empirical mean of confident test features. **We already provided a detailed analysis of α in Section 4.3 and Figure 2 (right) of the main paper**.

---

> > ### Comment · Reviewer_o7Us · 2025-08-01
> >
> > Thank you for the detailed response and the additional experimental results. This has clarified most of my concerns, though I would appreciate a follow-up on two points to ensure I fully understand the work.
> >
> > 1. Regarding W1, I appreciate the thorough explanation of the Gaussian assumption. However, my original question concerning the "a uniform class prior (line 134)" assumption appears to have been missed. While I understand this is an intuitive simplification, I am interested in understanding the method's robustness in settings where this prior doesn't hold, such as the imbalanced and non-stationary online scenarios mentioned in my review [1, 2, 3].
> > 2. For W2, the new results are compelling. To help me better situate the paper's contributions, could you please elaborate on the key distinctions from prior work? I am not challenging the paper's novelty, but rather seeking a clearer differentiation. Specifically, I had noted that "**The backpropagation-free TTA approach using pre-trained classifier weights as prototype anchors closely parallels [4]'s vision-only methodology...**" and would appreciate it if you could detail how your approach differs. Similarly, for my comment that "**the 'Knowledge banks' mechanism resembles those in [3, 5]... and ADAPT's update strategy... appears very similar,**" could you highlight the primary distinctions in your update strategy (Eq. 9)?

---

> ### Author Response · Authors · 2025-08-04
> **Response to Reviewer o7Us [1/2]**
>
> We sincerely appreciate the reviewer's thoughtful follow-up and their consideration of our previous explanations. We are pleased to provide our responses to the follow-up points below.
>
> > **W1: On the Robustness to the Uniform Class Prior Assumption in Non-Stationary.**
>
> **(1) Assumption on the Uniform Class Prior:**
>
> While a simplification, the uniform class prior is a **common assumption in source-free TTA [1,2,3]** where the true test distribution is unknown. Importantly, ADAPT is designed to mitigate this issue via its **fixed-size per-class knowledge bank**. This structure ensures class-conditional likelihoods are estimated from a **balanced set of historical samples**, providing inherent robustness to class imbalance. Our Q5 analysis further confirms this, showing reliable performance even with very few samples per class (e.g., 2-4).
>
> **References**:
>
> [1] A Hard-to-Beat Baseline for Training-free CLIP-based Adaptation (ICLR 2024)
>
> [2] Enhancing Zero-Shot Vision Models by Label-Free Prompt Distribution Learning and Bias Correcting (NeurIPS 2024)
>
> [3] Generalized Logit Adjustment: Calibrating Fine-tuned Models by Removing Label Bias in Foundation Models (NeurIPS 2023)
>
> **(2) Empirical Validation in Imbalanced and Non-Stationary Scenarios:**
>
> We acknowledge that the performance of our **source-free, training-free, and prompt-tuning-free** approach can be challenged when the test distribution undergoes continuous, abrupt shifts composed of low-confidence or distribution-shifted samples, as this makes the accurate estimation of a rapidly changing distribution inherently difficult.
>
> To position our work within the context of the literature the reviewer provided, we wish to highlight a **key distinction between two families of TTA protocols**. Our ADAPT aligns with one family [4, 5, 7], while the other [1, 2, 3] is built upon different experimental frameworks and evaluation goals. For this reason, a direct comparison with our work would not be equitable.
> Therefore, to explore ADAPT's behavior under the challenging conditions the reviewer mentioned, we have **designed an experiment that simulates an imbalanced and non-stationary scenario specifically within our TTA protocol**. While not a direct comparison to [1, 2, 3], we believe this analysis provides valuable insight into ADAPT's robustness.
>
> **Experimental Setup:**
> We partitioned 1,000 ImageNet classes into three non-overlapping sets (Group A, B, C) to randomly construct a 12,500-sample test stream for each phase. **Imbalance** was created by drawing **80%** samples from a single **"dominant" group** in each phase, with the remaining 20% from the other groups. **Non-stationarity** was induced by **abruptly shifting the dominant group** across four sequential phases: Phase 1 (A is dominant) → Phase 2 (B is dominant) → Phase 3 (C is dominant) → Phase 4 (A is dominant again).
>
> **Evaluation and Analysis:**
> We compared the zero-shot CLIP with ADAPT. The results, summarized in the table, highlight ADAPT's resilience across all phases:
>
> * **Rapid Initial Adaptation:** Quickly adapts in Phase 1 with a significant **+2.33** accuracy gain.
> * **Resilience to Abrupt Shifts:** When the distribution suddenly changes in Phase 2, ADAPT **not only adapts but improves, widening the gain to +2.89**. This showcases its ability to rapidly learn from new distributions without being destabilized by the shift.
> * **Cumulative Improvement:** In Phase 3, ADAPT achieves its peak gain of +6.01. This suggests that as **its knowledge bank is exposed to more diverse data, the model's adaptive capability becomes even more potent**.
> * **No Catastrophic Forgetting:** Most critically, when the stream **reverts to Group A in Phase 4,** ADAPT's performance remains high, achieving a robust gain of +3.76. This proves it can handle recurring distributions without catastrophically forgetting knowledge learned in earlier phases.
>
> In summary, this targeted experiment demonstrates that ADAPT **remains robust even under imbalanced and non-stationary test streams**. Its per-class knowledge bank enables consistent performance across shifting distributions, alleviating the limitations imposed by the uniform prior assumption. These findings strengthen the practical viability of ADAPT and support its application to real-world, dynamic environments.
>
> | Phase | Method | Group A | Group B | Group C | Avg. |
> | :--- | :--- | :--- | :--- | :--- | :--- |
> | Phase 1 (A) | CLIP | 68.60 | 66.02 | 64.41 | 67.92 |
> | | ADAPT (Online) | 70.99 (+2.39) | 66.50 (+0.48) | 68.03 (+3.62) | 70.25 (+2.33) |
> | Phase 2 (B) | CLIP | 69.65 | 66.38 | 65.22 | 66.59 |
> | | ADAPT (Online) | 73.16 (+3.51) | 68.97 (+2.59) | 69.87 (+4.65) | 69.48 (+2.89) |
> | Phase 3 (C) | CLIP | 66.35 | 66.21 | 64.91 | 65.18 |
> | | ADAPT (Online) | 71.46 (+5.11) | 70.46 (+4.25) | 71.25 (+6.34) | 71.19 (+6.01) |
> | Phase 4 (A) | CLIP | 68.90 | 67.05 | 64.37 | 68.26 |
> | | ADAPT (Online) | 72.52 (+3.62) | 71.38 (+4.33) | 68.68 (+4.31) | 72.02 (+3.76) |

---

> ### Author Response · Authors · 2025-08-04
> **Response to Reviewer o7Us [2/2]**
>
> > **W2: Detailed comparisons with prior works.**
>
> Thank you for this follow-up question. We are glad to clarify in detail the key distinctions from prior work.
>
> **(1) Distinction from T3A [4]**
>
> ADAPT and T3A differ fundamentally in their modeling paradigm, which directly impacts their robustness and applicability.
>
> **Methodology (Distribution-aware vs. Prototype-based):**
> * T3A is a **non-parametric** method using pre-trained classifier weights as prototype anchors. It adjusts these prototypes by computing **centroids** from a support set formed from a **batch of test data**. This centroid-based approach is **sensitive to outliers, especially for classes with very few confident samples**.
> * In contrast, ADAPT is a **parametric, distribution-aware** method. It models the class-conditional feature distribution ($\mu_k$ , $\Sigma$) from high-confidence historical samples. By representing each class with a global distribution rather than a simple centroid, the influence of any single outlier is naturally diminished.
>
> **Applicability (Online(bs=1) vs. Online(bs>1) ):**
> * T3A's reliance on **batch-based centroid** calculation makes it **unsuitable for strict online scenarios (bs=1)** where samples arrive sequentially.
> * ADAPT is explicitly designed to support **both transductive and online (bs≥1) settings**.
>
> **(2) Distinctions from DPCore [3] and DynaPrompt [5]**
>
> While these methods also use similar knowledge bank mechanisms, they are fundamentally **prompt-tuning methods**. In contrast, ADAPT is entirely **training-free and prompt-tuning-free**.
>
> **Stored Elements (Learnable Prompts vs. Raw Samples):**
> * The "banks" in **DPCore** and **DynaPrompt** store and continuously update a set of **learnable prompt vectors**, introducing **new parameters** that must be optimized.
> * ADAPT's knowledge bank is simpler: it stores the **raw feature vectors** of high-confidence historical test samples, **requiring no training or new parameters**.
>
> **Update Strategy:**
> * DPCore employs a **batch-based, distance-weighted update**, requiring **access to a new batch of test data**. This is incompatible with strict online (bs=1) settings.
> * DynaPrompt relies on computationally expensive **data augmentation** for its prompt selection and uses prompt-based **hard thresholds** to manage its buffer.
> * In contrast, our ADAPT (Eq. 9) is **batch-free, augmentation-free, and threshold-free**. It dynamically updates its knowledge bank based on the zero-shot prediction confidence of each sample, making it uniquely suited for true online adaptation.
>
> In summary, ADAPT distinguishes itself by being **a training-free, prompt-tuning-free, and distribution-aware method**. Its unique mechanisms lead to high efficiency and broad applicability in **both online (bs≥1) and transductive settings**. To clearly and intuitively illustrate the key distinctions between ADAPT and prior work, we provide a comparative summary in the table below.
>
> | Key Distinctions | ADAPT (Ours) | T3A [4] | DPCore [3] | DynaPrompt [5] |
> | :--- | :--- | :--- | :--- | :--- |
> | **Core Paradigm** | Distribution-Aware | Prototype-Based | Prompt-Tuning | Prompt-Tuning |
> | **Knowledge Bank Elements** | Confident Test Samples | - | Learnable Prompts, Statistics | Learnable Prompts |
> | **Training/Tuning Free** | ✓ | ✓ | ✗ | ✗ |
> | **Update Basis** | Sample-by-Sample | Batch-Based | Batch-Based | Sample-Wise Tuning |
> | **Key Dependencies** | - | Batch of Data | Batch of Data | Augmentation, Hard Thresholds |
> | **Applicability (Trans.)** | ✓ | ✓ | ✓ | ✓ |
> | **Applicability (Online, bs>1)** | ✓ | ✓ | ✓ | ✗ |
> | **Applicability (Online, bs=1)** | ✓ | ✗ | ✗ | ✓ |
>
> ---
>
> We hope this clarification resolves your concerns. If you have any further questions, we'd be glad to discuss them in more detail.

---

> ### Comment · Reviewer_o7Us · 2025-08-07
>
> Thank you for the detailed follow-up. The new experiments and clarifications have been very helpful in addressing my concerns.
>
> I believe my remaining confusion for both W1 and W2 stems from the specific definition of "**online setting**" used in your paper. Typically, in the TTA literature, the online setting is defined by a batch of data arriving at each time step, where the batch size is not strictly constrained (and "transductive setting" is the traditional "offline setting"). Your paper uses a more restrictive definition where the batch size is strictly 1, which could be referred to as "**online setting (bs=1)**." As the batch size is 1, the temporal correlation is different from the settings in [1-3].
>
> This is an important distinction for your comparison table. The table currently suggests T3A and DPCore are not applicable in an "online setting," which could confuse readers, as those papers state **their methods do work online**. My understanding is that they are applicable to conventional online settings (e.g., with a batch size of 16) but may not be designed for the bs=1 scenario. Clarifying this in the table—for example, by specifying the setting as "Online (bs=1)"—is important for positioning your work accurately. The table is very informative and will be a valuable addition to the paper with this clarification if included in the final version.
>
> I have one remaining question regarding W1: You mention a key distinction between "two families of TTA protocols" to explain why a direct comparison with some literature is not equitable. Could you please elaborate on what defines these two families and provide representative citations for each?

---

> > ### Author Response · Authors · 2025-08-08
> > **Response to Reviewer o7Us**
> >
> > Thank you very much for your insightful follow-up and for highlighting this important point regarding the definition of the *online setting* in test-time adaptation (TTA).
> >
> > ---
> >
> > We fully agree that the terminology surrounding online TTA can vary across the literature. To clarify our evaluation protocol, we wish to note that **the shift toward single-instance (batch size = 1) online TTA has become a widely accepted paradigm in recent literature**, especially following the emergence of efficient methods [1-7]. To avoid ambiguity, we will explicitly clarify in our paper that our evaluation setting follows the **strict, single-instance online protocol (batch size = 1)**, where test samples arrive one at a time and must be processed sequentially. Following the reviewer's advice, our main response above has been **updated** to reflect this clarification, including renaming the relevant row to "Applicability (Online, bs=1)".
> >
> >
> > To further clarify the distinction, we summarize the key differences between the two commonly adopted online TTA paradigms:
> >
> >
> > **1. Single-Instance Online TTA (Batch Size = 1)**
> >
> > This is the strict online scenario targeted by our method, **ADAPT**.
> >
> > * **Protocol:** Test samples are received and adapted to **one by one**, with **no access to future or batched samples**.
> > * **Adaptation Mechanism:** These methods typically **do not update the backbone network**. Instead, they leverage **lightweight plug-in modules** (e.g., prompts) to enhance predictions on a **per-sample basis**.
> > * **Representative Methods:** ADAPT (Ours), TPT [1], TDA [2], DPE [3], AWT [4], BCA [5] , DynaPrompt [6], RA-TTA [7].
> >
> >
> >
> > **2. Batch-Based Online TTA (Batch Size > 1)**
> >
> > A more relaxed online setting, assuming access to multiple test samples during adaptation.
> >
> > * **Protocol:** Adaptation is performed using **mini-batches of test data** (e.g., batch size = 64).
> > * **Adaptation Mechanism:** Mechanisms in this family are varied. Typically, some methods [8,9,10] **update the backbone network** (e.g., BatchNorm statistics) during testing by applying gradient-based losses (e.g., entropy minimization) or smoothing mechanisms (e.g., exponential moving average). Some cases, like T3A [11] and DPCore [12], refrain from direct backbone parameter updates but leverage batch data to construct statistics (e.g., prototypes) for inference.
> > * **Representative Methods:** TENT [8], CoTTA [9], RoTTA [10], T3A [11], DPCore [12].
> >
> >
> > **References**:
> >
> > [1] Test-Time Prompt Tuning for Zero-shot Generalization in Vision-Language Models (NeurIPS 2022)
> >
> > [2] Efficient Test-Time Adaptation of Vision-Language Models (CVPR 2024)
> >
> > [3] Dual Prototype Evolving for Test-Time Generalization of Vision-Language Models (NeurIPS 2024)
> >
> > [4] AWT: Transferring Vision-Language Models via Augmentation, Weighting, and Transportation (NeurIPS 2024)
> >
> > [5] Bayesian Test-Time Adaptation for Vision-Language Models (CVPR 2025)
> >
> > [6] DynaPrompt: Dynamic Test-Time Prompt Tuning (ICLR 2025)
> >
> > [7] RA-TTA: Retrieval-Augmented Test-Time Adaptation for Vision-Language Models (ICLR 2025)
> >
> > [8] Tent: Fully Test-time Adaptation by Entropy Minimization (ICLR 2021)
> >
> > [9] Continual Test-Time Adaptation (CVPR 2022)
> >
> > [10] Robust Test-Time Adaptation in Dynamic Scenarios (CVPR 2023)
> >
> > [11] Test-Time Classifier Adjustment Module for Model-Agnostic Domain Generalization (NeurIPS 2021)
> >
> > [12] DPCore: Dynamic Prompt Coreset for Continual Test-Time Adaptation (ICML 2025)
> >
> > ---
> >
> > We hope this clarification addresses your concern. If you have any further questions, we'd be glad to discuss them in more detail.

---

> > > ### Comment · Reviewer_o7Us · 2025-08-09
> > >
> > > Thank you for the follow-up clarification. Your explanation of the different TTA paradigms has resolved my remaining questions.
> > >
> > > I have just one final suggestion for the comparison table to make it even more comprehensive. You might consider adding an additional row for a setting like "Applicability (Online, bs>1)" or "Applicability (Online, larger bs)". This would explicitly show that while all the compared methods work in a general online setting, the unique strength of your method lies in its effectiveness under the more restrictive single-instance (bs=1) protocol. I believe including these new results and the detailed discussion from your rebuttal will further strengthen the paper.

---

> ### Author Response · Authors · 2025-08-09
> **Response to Reviewer o7Us**
>
> We sincerely thank you for this final suggestion. We completely agree that adding an additional row for "Applicability (Online, bs>1)" is a great way to highlight that ADAPT's unique strength lies in its **broad applicability and effectiveness** even under the more restrictive single-instance online (bs=1) protocol.
>
> We have already incorporated this change into the comparison table provided in our earlier response. The **updated table is also shown below** for added clarity, highlighting that DynaPrompt fails in the conventional online setting (bs>1) due to its instance-based model reset. We will also add this information and discussion to the final version of our paper, as it helps to further emphasize what was recognized as a key strength: "the framework's flexibility to be a major plus; its ability to operate in both online and transductive settings without requiring source data makes it broadly applicable."
>
> Thank you again for your constructive feedback throughout this process, which has significantly helped strengthen our paper.
>
>
> | Key Distinctions | ADAPT (Ours) | T3A [4] | DPCore [3] | DynaPrompt [5] |
> | :--- | :--- | :--- | :--- | :--- |
> | **Core Paradigm** | Distribution-Aware | Prototype-Based | Prompt-Tuning | Prompt-Tuning |
> | **Knowledge Bank Elements** | Confident Test Samples | - | Learnable Prompts, Statistics | Learnable Prompts |
> | **Training/Tuning Free** | ✓ | ✓ | ✗ | ✗ |
> | **Update Basis** | Sample-by-Sample | Batch-Based | Batch-Based | Sample-Wise Tuning |
> | **Key Dependencies** | - | Batch of Data | Batch of Data | Augmentation, Hard Thresholds |
> | **Applicability (Trans.)** | ✓ | ✓ | ✓ | ✓ |
> | **Applicability (Online, bs>1)** | ✓ | ✓ | ✓ | ✗ |
> | **Applicability (Online, bs=1)** | ✓ | ✗ | ✗ | ✓ |

---

### Official Review · Reviewer_c5tn · 2025-06-30

**Clarity:** 3
**Significance:** 2
**Originality:** 3
**Rating:** 4
**Confidence:** 3

**Summary:**

This paper proposes a Gaussian Discriminant Analysis (GDA)-based test-time adaptation (TTA) method for CLIP. The GDA framework is initialized with textual embeddings of prompted class names. At test time, it models visual features from the image encoder and yields predictions in a backpropagation-free manner. The evaluation includes commonly used benchmarks.

**Questions:**

1. The authors may need to compare ADAPT with previous memory-based methods under comparable prior knowledge from textual prompts to more convincingly demonstrate the superiority of the proposed algorithm.
2. Given that ADAPT ultimately adopts prompts from AWT [57], AWT may need also be compared and discussed.

**Ethical Concerns:**

["NO or VERY MINOR ethics concerns only"]

**Final Justification:**

My initial concern was that the manuscript does not clearly demonstrate raw performance improvements over previous backpropagation-free TTA methods, given that different prompts significantly influence the results of such methods, as shown in the authors' rebuttal.

Regarding raw performance improvements, the authors argue that recent backpropagation-free TTA methods have saturated on specific benchmarks, which partially supports the claim that the proposed method offers improvements. This reasoning is somewhat convincing.

Considering the strengths in other aspects of the work, I maintain my positive rating. I hope the authors can add the discussion on the impact of prompts and the saturating performance of backpropagation-free TTA methods, to help readers clearly understand the current state and challenges in this line of research.

**Limitations:**

yes

**Quality:**

4

**Strengths And Weaknesses:**

**Strengths**

1. The work is well-motivated, and the paper is well-written.
2. The method design appears principled and is supported by theoretical analysis.
3. The evaluation includes sufficient benchmarks, and the ablation studies are well-structured.

**Weaknesses**

The major concern lies in the exact performance advantage of the proposed method compared to existing backpropagation-free TTA baselines, especially memory-based TTA methods like TDA [18]. My understanding is that the main difference between ADAPT and memory-based methods is that memory-based approaches use a KNN-like strategy to model test-time visual features and yield predictions, while ADAPT uses a GDA-based approach to do the same. Since the text embeddings are fixed, both can incorporate prior knowledge from textual prompts. Although Tables 2 and 4 show performance gains of ADAPT over TDA, Table 7 indicates that ADAPT benefits significantly from GPT-generated prompts. To my knowledge, TDA employs hand-crafted prompts similar to the “CLIP Template” in Table 7. If we compare ADAPT and TDA using similar prompts, their performances seem comparable. This suggests that the observed performance gains in Tables 2 and 4 may largely result from the use of stronger GPT-generated prompts rather than from inherent advantages of the GDA-based method itself. Therefore, the advantage of the proposed GDA-based approach over memory-based methods remains uncertain based on the current manuscript.

---

> ### Author Rebuttal · Authors · 2025-07-29
>
> We thank the reviewer for their thoughtful comments and feedback.
>
> ---
> > **W1 & Q1: Clarification on the superiority of ADAPT over memory-based methods under comparable prompt conditions.**
>
> We would like to clarify that **the superior performance of ADAPT stems not from advanced prompt engineering, but from its core methodology**: distribution-aware adaptation.
>
> (1) **Methodological Advantage**: As the reviewer correctly noted, prior memory-based methods like TDA rely on a non-parametric, instance-wise k-NN approach, making their predictions highly sensitive to individual noisy samples in online scenarios. In contrast, ADAPT employs a parametric, GDA-based strategy to explicitly model the class-conditional distributions. This modeling not only reduces the influence of individual outliers but also leads to more stable and robust decision boundaries, as illustrated in Figure 3 of our main paper.
>
> (2) **Performance Beyond Prompt Engineering:** To directly address this concern, we evaluated ADAPT against TDA using a wide array of identical prompting strategies. The results summarized in the table below highlight that our ADAPT **consistently and significantly outperforms TDA across all tasks and prompting settings**. This confirms that **the gains stem from the method itself, not from stronger textual prompts**.
>
> (3) **Robustness to Prior Removal:** As further proof, Figure 2 (right) in our main paper shows that ADAPT maintains SOTA performance even when the prompt-based CLIP prior is **completely removed** ($\alpha=1.0$). This highlights that ADAPT’s strength lies in its data-driven, distributional modeling, not in external textual guidance.
>
> |  | Method | Vanilla | Ensemble | CLIP Template | GPT | GPT & Ensemble | GPT & CLIP Template |
> | :---: | :--- | :--- | :--- | :--- | :--- | :--- | :--- |
> | Task 1 | TDA | 63.92 | 66.01 | 65.93 | 65.92 | 65.94 | 65.96 |
> | | ADAPT (Online) | 64.70 (+0.78) | 66.56 (+0.55) | 66.51 (+0.58) | 66.53 (+0.61) | 66.57 (+0.63) | 66.58 (+0.62) |
> | | ADAPT (Trans.) | 65.53 (+1.61) | 67.16 (+1.15) | 67.23 (+1.30) | 67.09 (+1.17) | 67.10 (+1.16) | 67.12 (+1.16) |
> | Task 2 | TDA | 27.17 | 28.25 | 28.05 | 28.38 | 28.76 | 28.69 |
> | | ADAPT (Online) | 27.52 (+0.35) | 28.54 (+0.29) | 28.44 (+0.39) | 28.56 (+0.18) | 28.98 (+0.22) | 28.91 (+0.22) |
> | | ADAPT (Trans.) | 29.45 (+2.28) | 30.00 (+1.75) | 29.96 (+1.91) | 30.29 (+1.91) | 30.51 (+1.75) | 30.48 (+1.79) |
> | Task 3 | TDA | 66.92 | 67.39 | 64.46 | 69.62 | 69.04 | 69.18 |
> | | ADAPT (Online) | 67.43 (+0.51) | 67.74 (+0.35) | 67.62 (+3.16) | 70.76 (+1.14) | 69.95 (+0.91) | 69.90 (+0.72) |
> | | ADAPT (Trans.) | 69.55 (+2.63) | 70.00 (+2.61) | 70.38 (+5.92) | 72.43 (+2.81) | 72.05 (+3.01) | 71.91 (+2.73) |
>
> ---
> > **Q2: Comparison with AWT.**
>
> While ADAPT uses the same handcrafted prompts as AWT, their core adaptation strategies differ fundamentally. AWT aligns multiple augmented views of each test image with textual prompts via Optimal Transport, making its performance **highly dependent on visual augmentations**.
>
> To illustrate this, we compare ADAPT with both original  AWT and **a variant (AWT\*) without visual augmentations**. As shown in the table below, AWT suffers a **significant performance drop without augmentation**, confirming this reliance. In contrast, ADAPT achieves consistently strong performance without requiring such augmentations.
>
> This highlights ADAPT's ability to perform source-free, training-free adaptation without relying on computationally expensive visual augmentations. This makes ADAPT a more streamlined and versatile solution, which is readily applicable to both online and transductive settings.
>
> | Method | ImageNet | Aircraft | Clatech101 | Cars | DTD | EuroSAT | Flower102 | Food101 | Pets | Sun397 | UCF101 | Avg. |
> | :--- | :--- | :--- | :--- | :--- | :--- | :--- | :--- | :--- | :--- | :--- | :--- | :--- |
> | AWT | 71.32 | 29.22 | 95.54 | 69.93 | 55.56 | 58.61 | 75.07 | 85.54 | 92.53 | 70.58 | 72.51 | 70.58 |
> | AWT \* | 69.03 | 27.48 | 94.77 | 65.94 | 53.96 | 60.41 | 75.11 | 84.76 | 91.88 | 68.18 | 69.81 | 69.21 |
> | ADAPT (Online) | 70.91 | 28.95 | 94.48 | 68.19 | 55.20 | 68.19 | 75.56 | 83.81 | 92.01 | 70.57 | 70.66 | 70.78 |
> | ADAPT (Trans.) | 71.56 | 30.81 | 95.46 | 71.32 | 56.86 | 65.93 | 80.11 | 85.15 | 92.59 | 72.25 | 73.86 | 72.35 |

---

> > ### Comment · Reviewer_c5tn · 2025-08-05
> >
> > Dear Authors,
> >
> > Thank you for providing the detailed numbers, which are very helpful for identifying the performance gains of the proposed algorithm. The results appear to align with my concern: with equal prompts, ADAPT shows only limited improvements over TDA — approximately 0.6% on Task 1 datasets and 0.3% on Task 2 datasets in the online scenario. Moreover, TDA does not appear to have a higher computational burden than ADAPT.
> >
> > Given this, do the authors consider this a limitation in terms of both accuracy and efficiency?
> >
> >
> > Best regards,\
> > Reviewer c5tn

---

> ### Author Response · Authors · 2025-08-07
> **Response to Reviewer c5tn**
>
> We sincerely thank the reviewer for this thoughtful follow-up. While the raw margins (e.g., +0.6% on Task 1, +0.3% on Task 2) may appear small at first glance, they are in fact substantial given the saturation of these benchmarks and the nature of the shifts. Moreover, ADAPT’s design offers advantages in robustness, generality, and efficiency-flexibility that we believe outweigh the perceived limitations.
>
> ---
> **1. Significance in a Saturating Benchmark**
>
> Performance on the Task 1 and Task 2 benchmarks is approaching saturation, where the gap between top-tier methods is typically **less than 0.5%**, as shown **in the table** below. In this context:
> * **ADAPT achieves consistent, positive gains across all prompts**, for example, **+0.58%** on Task 1 (online) with CLIP templates, outperforming all listed recent SOTA works. Similarly, on the extremely challenging Task 2 (ImageNet-C, severity 5), **even a +0.3% gain** reflects improved robustness under the harshest corruption.
> * In the **transductive setting**, ADAPT’s advantage widens further (+1.30% on Task 1, +1.91% on Task 2), demonstrating its full potential when more test data is available.
> | Method | Venue | ImageNet | ImageNet-A | ImageNet-V | ImageNet-R | ImageNet-S | OOD Avg. | Avg. | Gains |
> | :--- | :--- | :--- | :--- | :--- | :--- | :--- | :--- | :--- | :--- |
> | TDA (CLIP Template) | CVPR 2024 | 69.23 | 63.19 | 65.06 | 80.67 | 51.51 | 65.11 | 65.93 | |
> | DMN-ZS | CVPR 2024 | 72.25 | 58.28 | 65.17 | 78.55 | 53.20 | 63.80 | 65.49 | -0.44 |
> | AWT | NeurIPS 2024 | 71.32 | 60.33 | 65.15 | 80.64 | 51.60 | 64.43 | 65.81 | -0.12 |
> | DPE | NeurIPS 2024 | 71.91 | 59.63 | 65.44 | 80.40 | 52.26 | 64.43 | 65.93 | +0.00 |
> | ZERO | NeurIPS 2024 | 71.17 | 62.75 | 65.23 | 80.75 | 50.59 | 64.83 | 66.10 | +0.17 |
> | BCA | CVPR 2025 | 70.22 | 61.14 | 64.90 | 80.72 | 50.87 | 64.41 | 65.57 | -0.36 |
> | ECALP | ICLR 2025 | 71.26 | 58.52 | 65.72 | 80.77 | 54.66 | 64.92 | 66.19 | +0.26 |
> | ADAPT (Online, CLIP Template) | | 70.37 | 63.72 | 65.15 | 80.95 | 52.34 | 65.54 | 66.51 | **+0.58** |
> | ADAPT (Trans., CLIP Template) | | 71.17 | 64.72 | 66.18 | 81.28 | 52.78 | 66.24 | 67.23 | **+1.30** |
> ---
> **2. Core Technical Insights**
>
> Beyond numerical improvements, ADAPT’s improvements come from a **shift from instance-based to distribution-aware adaptation**:
> * **TDA’s Limitation:** Purely instance-based, TDA relies on nearest-neighbor voting and is **sensitive to outliers and noisy samples**, leading to **unstable decision boundaries** under distribution shift (see **Figure 3** in the main paper).
> * **ADAPT’s Advantage:** By modeling **class-conditional distributions** (progressive mean estimation and shared covariance), **ADAPT stabilizes decision boundaries, naturally down-weights outliers**, and adapts more gracefully as shifts worsen.
> * **Hyperparameter Robustness:** Due to its instance-based nature, TDA is highly **sensitive to data distribution shifts and requires careful hyperparameter tuning**. As a result, its performance heavily depends on **multiple dataset-specific thresholds**. Using shared hyperparameters from ImageNet (noted as TDA\*) can lead to a significant performance **drop of 1.45%**. In contrast, **ADAPT uses the same set of hyperparameters across all datasets without per-dataset tuning**, making it much easier to deploy.
> * **Full-Data Utilization:** Unlike TDA, ADAPT can exploit the transductive setting to refine its estimates over the entire test distribution, unlocking further gains in accuracy and efficiency.
> | Method | ImageNet | ImageNet-A | ImageNet-V | ImageNet-R | ImageNet-S | OOD Avg. | Avg. | Gains |
> | :--- | :--- | :--- | :--- | :--- | :--- | :--- | :--- | :--- |
> | TDA (Online) | 69.23 | 63.19 | 65.06 | 80.67 | 51.51 | 65.11 | 65.93 | |
> | TDA\* (Online) | 69.23 | 63.19 | 62.55 | 79.15 | 48.28 | 63.29 | 64.48 | -1.45 |
> | ADAPT (Online) | 70.37 | 63.72 | 65.15 | 80.95 | 52.34 | 65.54 | 66.51 | +0.58 |
> | TDA (Trans.) | 69.25 | 63.00 | 65.13 | 80.43 | 51.55 | 65.03 | 65.87 | -0.06|
> | ADAPT (Trans.) | 71.17 | 64.72 | 66.18 | 81.28 | 52.78 | 66.24 | 67.23 | +1.30 |
> ---
> **3. Efficiency–Flexibility Advantage**
> * **Online Mode (High Accuracy):** On ImageNet, ADAPT’s online mode takes 1h 11m vs. TDA’s 50m, only +21 min for a **+1.4%** accuracy gain.
> * **Transductive Mode (High Efficiency):** ADAPT can process the full test set in 0.73 min, **\~70× faster** than TDA’s online mode, while maintaining strong performance.
> * **Unique Flexibility:** TDA’s purely instance-based online design cannot benefit from transductive processing, whereas ADAPT can operate in **either high-accuracy online** or **high-efficiency transductive** mode, offering a practical deployment advantage.
> ---
> In summary, ADAPT delivers SOTA performance on saturated benchmarks, improves robustness in severe shifts, requires **no dataset-specific tuning**, and offers a **flexible accuracy–efficiency trade-off** that TDA cannot match. These characteristics are core strengths, not limitations.

---

> > ### Comment · Reviewer_c5tn · 2025-08-07
> >
> > Dear Authors,
> >
> > Thank you for the detailed discussion. I believe it would be valuable for the community to incorporate these discussions into the paper, particularly regarding the impact of prompt engineering for backpropagation-free TTA deployment and the saturating benchmark. I will maintain my positive rating for this paper.
> >
> > All the best,\
> > Reviewer c5tn

---

> > > ### Author Response · Authors · 2025-08-08
> > > **Response to Reviewer c5tn**
> > >
> > > Dear Reviewer,
> > >
> > > Thank you for your thoughtful follow-up and positive feedback. We are very pleased to hear that our explanations have addressed your concerns.
> > >
> > > Sincerely,
> > >
> > > The Authors

---

### Official Review · Reviewer_Hyj2 · 2025-07-02

**Clarity:** 3
**Significance:** 3
**Originality:** 3
**Rating:** 5
**Confidence:** 5

**Summary:**

This paper introduces ADAPT, a novel backpropagation-free and distribution-aware test-time adaptation method designed for VLMs. By modeling class-conditional feature distributions via gaussian discriminant analysis and updating them with high-confidence test samples in a constructed knowledge bank, the method provides closed-form, training-free and efficient adaptation. ADAPT operates under both online and transductive settings and outperforms both backpropagation-based and training-free baselines across natural shifts, corruption robustness and fine-grained categorization tasks.

**Questions:**

1. What is the effect of significant noise in the early high-confidence predictions on long-tail adaptation?
2. What kind of historical samples or priors does ADAPT need?

**Ethical Concerns:**

["NO or VERY MINOR ethics concerns only"]

**Final Justification:**

All my concerns have been resolved, and I am now inclined to recommend acceptance of the paper.

**Limitations:**

Yes.

**Paper Formatting Concerns:**

There are no obvious issues with the paper's formatting. It would be advisable to slightly increase the line spacing between tables.

**Quality:**

3

**Strengths And Weaknesses:**

Strengths:

1. ADAPT avoids iterative updates or backpropagation at test time by formulating TTA as a probabilistic inference task with gaussian assumptions, enabling low-latency adaptation.
2. ADAPT explicitly estimates class-conditional feature distributions with a shared covariance matrix and uses knowledge banks for regularization.
3. ADAPT achieves SOTA performance on ImageNet variants and 10 fine-grained datasets under both online and transductive settings, while also being more efficient in time and memory usage.

Weaknesses:

1. While the Gaussian Discriminant Analysis formulation enables efficient closed-form inference, the assumption of unimodal Gaussian distributions with a shared covariance across classes may not capture the complexity of real-world data, especially in multimodal or highly structured feature spaces. This simplification can result in biased or suboptimal predictions in challenging domains (e.g., fine-grained classification or long-tail distributions), potentially limiting ADAPT's effectiveness in practical deployment scenarios.
2. The method depends on confidence thresholds, bank sizes and prior means from CLIP templates, which may require careful tuning.

---

> ### Author Rebuttal · Authors · 2025-07-29
>
> We thank the reviewer for the valuable feedback.
>
> ---
> > **W1: Assumption of Gaussian distributions with a shared covariance.**
>
> Thank you for this insightful question. We agree that the unimodal Gaussian assumption introduces a simplification and may be limiting in hypothetical scenarios involving highly complex feature distributions. However, as demonstrated in the main paper, **ADAPT consistently achieves SOTA performance across diverse and challenging settings**, including **domains with severe visual corruption (Table 3)** and **fine-grained classification tasks (Table 4)**. These empirical results suggest that, despite the assumption, our method remains robust and effective in real-world applications.
>
> To further investigate this design choice,  we conducted an ablation study where we **replaced the shared covariance matrix with class-wise covariance matrices**, keeping all other settings fixed. This led to a **notable drop in accuracy, along with increased memory usage and inference time**. We attribute this to data sparsity: in many TTA settings, only a few high-confidence samples are available per class, making class-wise covariance estimation unreliable and prone to overfitting.
>
> In contrast, our shared covariance pools statistics across classes, producing a more stable and generalizable estimate, which is crucial in online or data-scarce environments. This further supports our design choice: **the shared covariance assumption is not merely for reducing computational overhead but is crucial for improving stability and generalization**. As the reviewer also noted, a key strength of ADAPT lies in its broad applicability, particularly in practical TTA scenarios where low latency and limited resources are critical.
>
> | Method | Time | Acc | Gains | Mem. (GB) |
> | :--- | :--- | :--- | :--- | :--- |
> | CLIP | 8m | 66.74 | - | 0.79 |
> | ADAPT (Online) | 1h 11m | 70.91 | +4.17 | 0.93 |
> | w/ Separate Cov. | 1h 44m | 53.80 | -12.94 | 2.89 |
> | ADAPT (Trans.) | 0.73m | 71.56 | +4.82 | 3.37 |
> | w/ Separate Cov. | 1.40m | 66.16 | -0.58 | 4.62 |
>
> ---
> > **W2: Hyperparameter analysis.**
>
> We already provided a detailed hyperparameter analysis in **Section 4.3** of our main paper, particularly in **Figure 2 and Table 7**. For clarity, we summarize our response as follows:
>
> (1) **Confidence Threshold:** Our method **does not use any manually defined confidence threshold**. Instead, we adopt a fixed-size memory bank that is dynamically updated; new samples with higher confidence scores replace those with the lowest scores. This adaptive mechanism eliminates the need for hand-crafted thresholds and ensures that only the most reliable samples are retained.
>
> (2) **Bank Size (L):** ADAPT exhibits strong robustness to this parameter. As shown in **Figure 2 (left)**, performance in the transductive setting remains stable across a wide range of values for L, while in the online setting, accuracy plateaus once L exceeds 10. We thus use a single, fixed configuration across all datasets (L=16 for online, L=6 for transductive), **with no per-dataset tuning required**.
>
> (3) **CLIP Prior Means:** ADAPT is not sensitive to the initialization of the CLIP prior means. **Table 7** verifies the robustness of our method under various mean settings. Furthermore, **Figure 2 (right)** shows that **even when the CLIP prior mean is entirely removed (α=1.0), ADAPT still achieves state-of-the-art performance**.
>
> In summary, **ADAPT does not rely on dataset-specific hyperparameter tuning or static priors**. Instead, it leverages dynamic, data-driven mechanisms to adaptively maintain a reliable knowledge bank. This design choice ensures strong performance, high robustness, and ease of deployment in real-world settings.
>
> ---
> > **Q1: What is the effect of significant noise in the early high-confidence predictions on long-tail adaptation?**
>
> To assess the impact of early-stage prediction noise under long-tailed distributions, we conducted a controlled experiment with the following setup:
>
> (1) **Long-tail dataset construction**: We constructed a **long-tailed subset from ImageNet-1K by sampling classes according to a power-law distribution**, resulting in:
>
> * **Head class**: 61 classes (≥40 shots, 2,719 images)
> * **Medium class**: 338 classes (10–40 shots, 7,396 images)
> * **Tail class**: 601 classes (≤10 shots, 2,416 images)
>
> (2) **Worst-case noise injection**: To simulate an extreme scenario, we deliberately injected noise during the **early 30% phase of adaptation**. Specifically, we **corrupted 30% of the high-confidence predictions by randomly reassigning them to incorrect classes**. This models a challenging case where early memory is contaminated with misleading information.
>
> (3) **Evaluation and Findings**: We compared the baseline CLIP model with ADAPT in this adverse setting. The results, summarized in the table below, yield three key observations:
>
> * While both methods degrade under heavy noise, **ADAPT consistently achieves higher final accuracy (64.27% vs. 63.25%)**, demonstrating superior overall robustness.
>
> * For the **extremely few-shot tail classes**, both methods are highly vulnerable to the strong initial misguidance, exhibiting a comparable **performance drop**. This is an expected outcome, as a 30% initial corruption can heavily skew parameter estimates when very few samples are available.
>
> * ADAPT demonstrates **self-correction capability**: As adaptation progresses, the accumulation of correctly identified samples (especially from head/medium classes) allows the model to progressively refine its distribution estimates, effectively mitigating the influence of the initial noisy predictions. **This confirms that ADAPT is a more reliable solution for practical, noisy, long-tailed scenarios**.
>
> | Method | Head class | Med class | Tail class | Avg. |
> | :--- | :--- | :--- | :--- | :--- |
> | CLIP | 68.44 | 69.69 | 68.42 | 69.33 |
> | w/ Noise | 62.16 (-6.28) | 64.03 (-5.66) | 62.09 (-6.33) | 63.25 (-6.08) |
> | ADAPT (Online) | 69.22 | 70.31 | 69.08 | 69.83 |
> | w/ Noise | 63.55 (-5.67) | 65.20 (-5.11) | 62.25 (-6.83) | 64.27 (-5.56) |
> ---
>  > **Q2: Clarification on the required priors and historical samples for ADAPT.**
>
> ADAPT operates in a **training-free and label-free** manner. The only information it uses is derived from (1) a small set of high-confidence samples from the test stream itself and (2) optional class-mean priors from CLIP.
>
> (1) **Historical Samples:** ADAPT maintains a fixed-size memory bank containing previously seen test samples with high-confidence predictions. These are not labeled samples, but are **selected based on prediction confidence during inference**. The memory bank is **continuously updated, replacing the lowest-confidence entries with more reliable samples**, and is used to guide the adaptation of incoming samples. This strategy eliminates the need for manual thresholds, making ADAPT a fully self-supervised approach.
>
> (2) **CLIP Prior (Optional):** We optionally initialize the class-wise means using CLIP’s textual priors, but this prior is not essential. As shown in **Table 7** and **Figure 2** of the main paper, **even when these CLIP priors are removed (α = 1.0), ADAPT still achieves strong performance**. This highlights the robustness of our approach to the choice of prior.
>
> In essence, ADAPT is a self-contained method that adapts directly to the test stream, eliminating the need for external labeled data or strong, fixed priors.

---

> > ### Comment · Reviewer_Hyj2 · 2025-08-06
> >
> > Thank you for your response. All my concerns have been resolved, and I am now inclined to recommend acceptance of the paper.

---

> ### Author Response · Authors · 2025-08-07
> **Response to Reviewer Hyj2**
>
> Dear Reviewer,
>
> Thank you very much for your thoughtful follow-up and positive feedback. We're glad that our responses have fully addressed your concerns, and we truly appreciate your decision to recommend acceptance of our work.
>
> Sincerely,
>
> The Authors

---

### Official Review · Reviewer_uiWy · 2025-07-02

**Clarity:** 3
**Significance:** 3
**Originality:** 3
**Rating:** 4
**Confidence:** 4

**Summary:**

The paper proposes a gradient free method for adapting vision–language models at test time. The prediction is treated as Gaussian mixture class assignment, where the parameters of each class are updated based on a set of highly confident representatives of that class.
The paper treats two settings: (i) online: as each test sample arrives, its embedding is scored, and if confident, added to the per-class bank. Then, the class Gaussian parameters are recomputed in closed form before making that sample’s prediction. (ii) once the entire unlabeled test set is available, the method builds the banks from the most confident samples.
This allows the predictions, modeled as GMM assignment, to adapt to test time distribution changes.

**Questions:**

- Have the authors considered a bank-pruning step (periodically removing samples whose likelihood falls below a threshold)? This can help addressing some of the weaknesses.
- The shared covariance assumption seems very restrictive. Have the authors considered experimenting with separate covariance matrices, even on a small scale example?

**Ethical Concerns:**

["NO or VERY MINOR ethics concerns only"]

**Final Justification:**

The authors successfully addressed my concerns, and accordingly I am raising my score.

**Limitations:**

See weaknesses above.

**Quality:**

3

**Strengths And Weaknesses:**

**Strengths:**
The paper is well written and it proposes a simple and clean solution to test time adaptations. The empirical evaluation is extensive and convincing.

**Weaknesses:**
- The memory banks, build from high confidence samples, are central to the method’s efficiency, but they also introduce a potential failure, especially if significant shift occurs. If the initial CLIP zero-shot predictions are substantially misaligned with the new domain, their confidence scores will become unreliable, causing the memory banks to be built from misclassified examples. This, in turn, will corrupt the estimated class means and shared covariance.
- Similarly, the question of how having classes with very few high-confidence samples affects the estimation of the parameters, and therefore the results.
-  The paper offers no formal guarantees on how quickly the estimated parameters converge, or bound on the divergence from the true distribution.
- Evaluation relies exclusively on CLIP’s embedding space.  It’s unclear how well its assumptions and mechanisms transfer to other models.
- The method may be  highly dependent on the randomized order of samples in the online case. However, no standard error or other form of confidence is reported for repetitions of each experiment.

Despite these limitations, the paper offers a simple, elegant solution to a significant problem. I would be happy to revisit my evaluation—and potentially raise my score, if the authors address the identified weaknesses, particularly by reporting standard deviations or confidence intervals for their results.

---

> ### Author Rebuttal · Authors · 2025-07-28
>
> We thank the reviewer for the constructive comments. We provide our feedback as follows.
>
> > **W1: Performance and robustness under significant shifts.**
>
>  We agree that robustness against unreliable predictions under severe domain shift is essential. To further assess our method under significant shifts with synthetic corruptions, we benchmarked ADAPT against TDA, a state-of-the-art memory-based method, **across five levels of increasing shift severity**. As shown in the table below, while all methods degrade as the shift intensifies (an expected trend in an unsupervised setting), **our ADAPT consistently outperforms TDA across all levels**.
>
> Notably, ADAPT's performance advantage is even more significant under severe shifts (e.g., Level 4 and Level 5), where the risk of memory corruption is greatest. This provides strong evidence that our proposed mechanism is more resilient to noisy pseudo-labels, validating **its superior robustness for practical, severe distribution shifts**.
>
> | Shift Level | 1 | 2 | 3 | 4 | 5 |
> | :--- | :--- | :--- | :--- | :--- | :--- |
> | CLIP | 59.68 | 52.97 | 46.79 | 37.51 | 25.50 |
> | TDA | 60.42 | 54.15 | 48.35 | 36.84 | 28.34 |
> | ADAPT (Online) | 61.37 | 54.70 | 48.61 | 39.28 | 28.56 |
> | ADAPT (Trans.) | 62.04 | 55.51 | 49.68 | 40.76 | 30.29 |
>
> ---
> > **W2: How having classes with very few high-confidence samples affects the results.**
>
> We analyze how the number of high-confidence samples per class (denoted as **N**, ranging from 2 to 10) influences performance. As shown in the table, while **our online method** is somewhat sensitive to extremely low sample counts (e.g., 2 per class), it recovers quickly and achieves robust performance  **with only 4 to 6 samples per class**.
>
> Notably, **our transductive ADAPT** maintains high and stable performance  **even with just 2 high-confidence samples per class**. We attribute this to its ability to leverage the global structure of the full test batch, which regularizes parameter estimates and ensures robustness under data scarcity.
>
> |  | N | 2 | 4 | 6 | 8 | 10 |
> | :--- | :--- | :--- | :--- | :--- | :--- | :--- |
> | Online | Task 1 | 59.61 | 64.19 | 65.50 | 66.10 | 66.33 |
> | | Task 2 | 22.95 | 26.00 | 27.26 | 27.93 | 28.22 |
> | | Task 3 | 63.87 | 67.46 | 69.04 | 69.89 | 70.31 |
> | Trans. | Task 1 | 67.02 | 67.11 | 67.09 | 67.10 | 67.04 |
> | | Task 2 | 30.26 | 30.29 | 30.29 | 30.23 | 30.23 |
> | | Task 3 | 72.15 | 72.21 | 72.43 | 72.24 | 72.25 |
>
> ---
> > **W3: On the convergence and divergence bounds of estimated parameters.**
>
> We appreciate the reviewer’s insightful question. Providing formal guarantees for convergence is inherently difficult, as our regularization term $\mathcal{R}(z_i;\mathcal{B})$ relies on a knowledge bank and introduces non-decomposable dependencies. This type of regularizer poses fundamental challenges, and to the best of our knowledge, establishing general convergence theory for such structures remains an open problem in the broader learning theory community.
>
>
> Given these theoretical challenges, we instead provide a thorough empirical validation of our parameter estimation. We **conducted experiments on 100 randomly selected ImageNet classes to measure the divergence between our estimated distribution parameters ($\mu_k$ , $\Sigma$) and those of the ground-truth empirical distribution**.
>
> The results, averaged over **10 runs** and summarized below, demonstrate the high accuracy of our estimations. As the table shows, the estimation errors are consistently low and stable. The small Frobenius distances (F-dist) for the mean ($\mu$) and covariance ($\Sigma$), along with the low overall error, confirm that the estimated distribution closely aligns with the ground-truth. This strong empirical evidence demonstrates that, while a formal proof is an open challenge, our method effectively converges to the true underlying data distribution in practice. It achieves this highly accurate and robust estimation without requiring any supervision.
>
> | F-dist μ | F-dist Σ | Error |
> | :---: | :---: | :---: |
> | 0.1466 ±0.0055 | 0.2123 ±0.0003 | 0.2897 ±0.0174 |
>
> ---
>  > **W4: Evaluation with different VLMs.**
>
> We have already evaluated ADAPT with different VLMs in **Appendix B.3**. For reference, we present the summary table below, and more detailed results can be found in **Figure 3 of the Appendix**. ADAPT delivers consistent gains across various models, including **CLIP (ViT-B/16, RN50)**, **BLIP** [1], and **ALBEF** [2]. This demonstrates that ADAPT is model-agnostic and broadly applicable across different VLM architectures.
>
> | Method | Task 1 | Task 2 | Task 3 |
> | :--- | :--- | :--- | :--- |
> | **BLIP** [1] | 46.04 | 22.06 | 49.81 |
> | w/ ADAPT (Online) | 56.67 (+10.63) | 26.70 (+4.64) | 62.23 (+12.42) |
> | w/ ADAPT (Trans.) | 57.59 (+11.55) | 27.48 (+5.42) | 62.41 (+12.60) |
> | **ALBEF** [2] | 33.88 | 15.21 | 31.16 |
> | w/ ADAPT (Online) | 40.27 (+6.39) | 19.79 (+4.58) | 43.34 (+12.18) |
> | w/ ADAPT (Trans.) | 43.26 (+9.38) | 20.79 (+5.58) | 45.78 (+14.62) |
>
> [1] BLIP: Bootstrapping Language-Image Pre-training for Unified Vision-Language Understanding and Generation (ICML 2022)
>
> [2] Align Before Fuse: Vision and Language Representation Learning with Momentum Distillation (NeurIPS 2021)
>
> ---
> > **W5: Report the standard deviation of the results.**
>
> To address the concern regarding stability, we evaluated our method **over 10 random permutations** of the test stream, reporting the average performance and standard deviation across **three tasks**. The results confirm our method's high stability, with a **consistently low standard deviation** across all tasks. These standard deviations will be included in the tables of our final manuscript to formally report the method's reliability.
>
> Task1:
> | ImageNet | ImageNet-A | ImageNet-V | ImageNet-R | ImageNet-S | OOD Avg. | Avg. |
> | :--- | :--- | :--- | :--- | :--- | :--- | :--- |
> | 70.72 ±0.10 | 63.45 ±0.21 | 64.56 ±0.16 | 80.55 ±0.09 | 53.08 ±0.09 | 65.41 ±0.06 | 66.47 ±0.06 |
>
> Task2:
> | Gauss. | Shot | Impul. | Defoc. | Glass | Motion | Zoom | Snow | Frost | Fog | Brig. | Contr. | Elastic | Pixel | JPEG | Avg. |
> |  :--- | :--- | :--- | :--- | :--- | :--- | :--- | :--- | :-- | :--- | :--- | :--- | :--- | :-- | :--- | :--- |
> | 15.71 ±0.05 | 16.82 ±0.05 | 15.81 ±0.07 | 26.34 ±0.06 | 17.90 ±0.07 | 27.26 ±0.09 | 25.42 ±0.08 | 36.25 ±0.07 | 34.66 ±0.07 | 40.89 ±0.06 | 60.28 ±0.06 | 19.84 ±0.22 | 16.10 ±0.06 | 37.30 ±0.10 | 37.23 ±0.08 | 28.52 ±0.03 |
>
> Task3:
> | Aircraft | Caltech101 | Cars | DTD | EuroSAT | Flower | Food101 | Pets | Sun397 | UCF101 | Avg. |
> |:--- | :--- | :--- | :--- | :--- | :--- | :--- | :--- | :--- | :--- | :--- |
> | 28.93 ±0.21 | 94.60 ±0.15 | 68.05 ±0.27 | 53.87 ±0.61 | 67.24 ±0.71 | 76.28 ±0.34 | 83.91 ±0.04 | 90.37 ±0.58 | 71.26 ±0.26 | 71.82 ±0.45 | 70.64 ±0.10 |
>
> ---
> > **Q1: Experiment with the bank-pruning step.**
>
> Thank you for this constructive suggestion. We implemented a bank-pruning mechanism as a new ablation study. We **periodically (every 1k steps) remove a portion of the lowest-confidence samples (e.g., 10%, 30%, 50%) from the memory bank**. This pruning is done on a per-class basis, as a single global confidence threshold can be unfair to naturally 'harder' classes. The results, presented below, show that this pruning strategy consistently degrades performance, especially on the more challenging Image-Net-A dataset with distribution shift.
>
> This outcome suggests that **ADAPT inherently performs a form of soft dynamic filtering, effectively mitigating the impact of noisy samples without requiring explicit pruning**. Through its regularized parameter estimation process, the method naturally down-weights the influence of unreliable samples rather than discarding them entirely. Introducing an explicit or aggressive pruning mechanism may, in fact, be counterproductive, as it risks eliminating samples that could become informative as adaptation progresses. This finding highlights the robustness and stability of our original design, which avoids premature decisions while remaining resilient to noise.
>
> | | ImageNet | ImageNet-A |
> | :--- | :--- | :--- |
> | ADAPT (Online) | 70.91 | 63.32 |
> | w/ 0.1 pruning | 69.16 | 50.69 |
> | w/ 0.3 pruning | 69.03 | 50.60 |
> | w/ 0.5 pruning | 69.02 | 50.56 |
> | w/ 0.7 pruning | 68.97 | 50.52 |
>
>
> ---
> > **Q2: Using separate covariance matrices.**
>
> As we detail in Appendix A.4 (Discussion), we provide strong empirical evidence that the class-conditional features approximately follow Gaussian distributions with a shared covariance matrix.
>
> To further compare with class-wise separate covariance matrices, we conducted additional experiments on ImageNet by **replacing our shared covariance with class-specific covariance matrices**, keeping all other settings identical. The results, summarized below, show that this **leads to a significant drop in accuracy, alongside a sharp increase in computation time and memory usage**.
>
> We attribute this performance degradation to data sparsity. **Estimating a full covariance matrix for each class becomes unreliable with few high-confidence samples, leading to poor generalization and overfitting**. In contrast, the shared covariance pools information across all classes, resulting in a much more stable and robust estimate, which is critical in online or data-scarce settings.
>
> Ultimately, this ablation confirms that **our shared covariance assumption is not a limitation, but a crucial design choice**. It strikes an effective trade-off between accuracy, robustness, and computational efficiency, making our method practical for real-world, resource-constrained scenarios.
>
> | Method | Time | Acc | Gains | Mem. (GB) |
> | :--- | :--- | :--- | :--- | :--- |
> | CLIP | 8m | 66.74 | - | 0.79 |
> | ADAPT (Online) | 1h 11m | 70.91 | +4.17 | 0.93 |
> | w/ Separate Cov. | 1h 44m | 53.80 | -12.94 | 2.89 |
> | ADAPT (Trans.) | 0.73m | 71.56 | +4.82 | 3.37 |
> | w/ Separate Cov. | 1.40m | 66.16 | -0.58 | 4.62 |

---

> > ### Comment · Reviewer_uiWy · 2025-08-01
> >
> > Dear authors,
> >
> > Thank you for addressing my concerns in W2, W4 and W5.
> > I appreciate the effort you have done with regard to W3, although my concern remains. Nevertheless, I consider this minor compared to other concerns that I had.
> >
> > I believe the inclusion of your analysis in response to Q2 in the appendix can improve the paper.
> >
> > My remaining questions are with respect to **W1 and Q1**:
> > Could you provide details on the shifts you tested for? How do you explain your relative advantage being larger for severe shifts while relying mainly on representative examples? Do the shifts you tested for affect the distribution mainly around modes or in tails?
> > The same questions relate also to the negative effect of pruning (Q1) that you observed.

---

> ### Author Response · Authors · 2025-08-04
> **Response to Reviewer uiWy**
>
> We sincerely appreciate the reviewer's thoughtful follow-up and their consideration of our previous explanations. We agree that the analysis provided in response to Q2 offers valuable context and will include it in the appendix of the revised version. Regarding W1 and Q1, we are pleased to provide additional details and clarifications below.
>
> ---
>
> > **W1 & Q1: On Shift Robustness and the Effect of Pruning**
>
> **(1) Nature of Shifts and ADAPT's Growing Advantage (W1):**
>
> **(1-1) Nature of the Tested Shifts (ImageNet-C)**:
>
> In W1, we follow the common protocol of evaluating on the **corruption-shift benchmark ImageNet-C [1]** to test robustness against challenging domain shifts.  This benchmark introduces **15 corruption types** (e.g., Gaussian noise, blur, JPEG compression) **across 5 severity levels**. These corruptions impact **both the central (mode) and peripheral (tail) regions of the feature distribution**. For instance, blurring and defocus affect core object representations, while noise and distortions disproportionately degrade rare or ambiguous samples. This results in a progressively more severe distribution shift across all severity levels.
>
> **(1-2) Rationale for ADAPT's Growing Advantage on Severe Shifts**:
>
> The key lies in the **contrast between CLIP's static design** and **ADAPT's dynamic adaptation**:
>
> * **CLIP's Sharp Decline**: CLIP uses a fixed classifier calibrated on clean data, making it highly sensitive to corrupted or off-distribution features. As corruption severity increases, CLIP's predictions become misaligned with the target test distribution, leading to **sharp performance degradation**.
>
> * **TDA's Vulnerability**: TDA, as a memory-based method, slightly improves upon CLIP in moderate shifts. However, its **k-NN-like, instance-based** strategy makes it highly vulnerable to outliers in noisy conditions. This core weakness is exposed under severe corruption, causing its performance to dramatically degrade and even drop below the static CLIP at Level 4 (36.84 vs 37.51).
>
> * **ADAPT's Resilience**: In contrast, ADAPT **dynamically estimates class-wise distributions** using soft pseudo-labels and confidence-based memory banks. While it relies on representative examples, ADAPT does **not rely on perfect pseudo-labels**. Instead, its use of soft pseudo-labeling and fixed-size class-wise memory banks **prevents dominance by corrupted outliers** and allows it to **progressively model the distribution of the corrupted data**. This soft adaptation avoids overfitting to outliers while gradually aligning with the shifted data.
>
> * **Advantage is Relative**: Because ADAPT degrades more gracefully than CLIP, the **relative performance gap** grows with increasing corruption severity. As shown in our W1 results, ADAPT (Trans.) outperforms CLIP by **+2.36%** at Level 1, growing to **+4.79%** at Level 5.
>
> ---
>
> **(2) Negative Effect of Pruning (Q1):**
>
> **(2-1) Nature of the Tested Shifts (ImageNet-A)**:
>
> In our Q1 experiment, we consider both **ImageNet** and **ImageNet-A**. The latter represents a **natural distribution shift**, consisting of naturally occurring, real-world examples that models find inherently difficult, thus representing the long tail of the data distribution.
>
> **(2-2) Why Hard Pruning Fails**:
>
> Hard pruning removes samples based solely on confidence. However, under severe shifts like ImageNet-A, many **samples with relatively low confidence remain semantically valuable**. Discarding them reduces diversity and limits the model's ability to adapt to unfamiliar patterns in the test data.
>
> Instead of pruning, ADAPT **dynamically updates its class-wise knowledge bank** and applies **soft confidence-based weighting**. This ensures that even samples with relatively low confidence can contribute proportionally if they are informative. This allows ADAPT to gradually refine its distribution estimates using the full test stream, balancing robustness and adaptability even in low-data or high-noise regimes.
>
> **References**:
>
> [1] Benchmarking Neural Network Robustness to Common Corruptions and Perturbations (ICLR 2019)
>
> ---
>
> We hope this clarification resolves your concerns. If you have any further questions, we'd be glad to discuss them in more detail.

---

> > ### Comment · Reviewer_uiWy · 2025-08-04
> > **Response**
> >
> > Dear authors,
> >
> > Thank you for your detailed response.
> >
> > I found your explanations convincing, and I strongly recommend incorporating these additional results and discussions in your paper, including the pruning one. I believe it provides a needed perspective on why your method works.
> >
> > Accordingly, I am planning to raise my score.

---

> > > ### Author Response · Authors · 2025-08-05
> > > **Response to Reviewer uiWy**
> > >
> > > Dear Reviewer,
> > >
> > > Thank you very much for your positive feedback and support. We are pleased to hear that you found our explanations convincing and truly appreciate your decision to raise the score.
> > >
> > > As you suggested, we will incorporate the additional results and discussions, including the pruning analysis, into the final version of the manuscript. Thank you again for your thoughtful follow-up and valuable guidance throughout the review process.
> > >
> > > Sincerely,
> > > The Authors

---

### Comment · Reviewer_o7Us · 2026-04-30
**Camera-Ready Version Missing Several Commitments Made During Rebuttal**

Dear Authors,

Congratulations on the acceptance of this paper. Having now had the opportunity to compare the camera-ready version against the rebuttal commitments made during the review process, I would like to raise a concern that I believe warrants clarification, both for transparency to the community and for potential incorporation into a future arXiv revision.

During the rebuttal, the authors made several explicit commitments to incorporate specific analyses and experiments into the final manuscript. While I appreciate that a number of these were indeed added (e.g., the separate-covariance ablation in Table 11, the severity-level evaluation in Table 16, the low-confidence sample analysis in Table 17, and the extended baselines in Tables 2 and 4), several substantive commitments — including some that were central to reviewers' decisions to support acceptance — appear to be absent from the camera-ready version. To name the most important one per reviewer:

**Reviewer o7Us**: The detailed comparison table differentiating ADAPT from T3A, DPCore, and DynaPrompt — including the "Applicability (Online, bs=1)" vs. "Applicability (Online, bs>1)" rows — which the authors explicitly stated "We will also add this information and discussion to the final version of our paper." This was central to addressing the reviewer's concerns about novelty positioning, and was acknowledged by the reviewer as a key contribution to the paper. It does not appear in the camera-ready version.

**Reviewer uiWy**: The standard deviations across 10 random permutations of the test stream, which the authors stated "will be included in the tables of our final manuscript to formally report the method's reliability." None of the tables in the camera-ready report standard deviations or confidence intervals. The bank-pruning ablation (Q1) — which the reviewer specifically highlighted with "I strongly recommend incorporating these additional results and discussions in your paper, including the pruning one" and to which the authors agreed — is also absent.

**Reviewer c5tn**: The prompt-controlled comparison between ADAPT and TDA across six prompting strategies, the discussion of saturating benchmark performance, and the AWT/AWT* augmentation comparison. The reviewer explicitly wrote "I hope the authors can add the discussion on the impact of prompts and the saturating benchmark, to help readers clearly understand the current state and challenges in this line of research." None of these discussions appears in the camera-ready version, despite this being central to the reviewer's stated reason for maintaining a positive rating.

**Reviewer Hyj2**: The long-tail adaptation experiment with worst-case noise injection (the Head/Medium/Tail-class evaluation), which was the principal new analysis requested by this reviewer. It is not in the camera-ready version.

The above are only the single most prominent omission per reviewer; several additional commitments (e.g., the imbalanced/non-stationary phase experiment, the empirical convergence analysis with Frobenius distances, the "two families of TTA protocols" discussion) are similarly missing.

I recognize that page-limit constraints can sometimes preclude full incorporation of every rebuttal addition. However, the volume and significance of the omitted material — particularly content that reviewers explicitly cited as the basis for their support — suggests this is more than a routine trimming. At minimum, I would encourage the authors to:

Incorporate these analyses into the arXiv version, where page limits do not apply;
Acknowledge in the paper or the supplementary which rebuttal-promised content was deferred and where it can be found;
Confirm publicly here whether these additions are planned.

I raise this in good faith, in the spirit of the responsible reviewing initiative and the broader goal of preserving the integrity of the rebuttal process. Written commitments that demonstrably influenced reviewer scores are not stylistic preferences to be revisited at the authors' discretion after acceptance — honoring them is a basic expectation of academic integrity, and reviewer scores given on the basis of commitments that are subsequently dropped are effectively obtained under false pretenses.

Thank you for considering this.

---

> ### Public Comment · ~Sungeun_Hong1 · 2026-05-04
> **Clarification on Rebuttal Additions**
>
> Dear Reviewer,
>
> Thank you very much for raising this concern and for your continued attention to our work. We sincerely apologize that some rebuttal-related additions were not sufficiently visible in the NeurIPS camera-ready version.
> We would like to clarify that these items were not intentionally omitted, nor did we intend to disregard the commitments made during the rebuttal. During the preparation of the final manuscript, we had to balance completeness, readability, and the strict page constraints of the NeurIPS camera-ready version. We also considered that the NeurIPS rebuttal discussions on OpenReview are publicly accessible, allowing interested readers to refer to our detailed reviewer-specific responses for additional analyses and clarifications. As a result, we prioritized several core additions that directly supported the main narrative of the paper, including the separate-covariance ablation in Table 11, the severity-level evaluation in Table 16, the low-confidence sample analysis in Table 17, and the extended baselines in Tables 2 and 4.
> That said, we fully agree that relying on the rebuttal discussion alone was not sufficient, especially for analyses and discussions that reviewers explicitly requested and considered important. We should have made these additions easier to locate in the paper or supplementary material.
> To address this, we have made an arXiv version and supplementary material available at: https://arxiv.org/abs/2508.15568
> In particular, the arXiv version and supplementary material include additional analyses and discussions related to the rebuttal responses, including:
> (1) a comparison table and discussion differentiating ADAPT from T3A, DPCore, and DynaPrompt, including different online applicability settings, to clarify the relationship between different TTA protocol families (Table 12);
> (2) standard deviations across random test-stream permutations, together with the bank-pruning ablation and discussion (Tables 13 and 22);
> (3) a prompt-controlled comparison with TDA and a comparison with AWT/AWT* (Tables 18 and 19);
> (4) a long-tailed adaptation experiment with worst-case noise injection and Head/Medium/Tail-class evaluation (Table 23).
> We sincerely appreciate the reviewer’s good-faith concern. We apologize again for not making all rebuttal-related additions sufficiently visible in the camera-ready version, and we hope that the arXiv version and supplementary material address the concerns raised here.
>
> Thank you again for helping us improve the transparency and completeness of the paper.

---

### Note · Authors · 2025-08-12

Dear Program Chair, Senior Area Chair, Area Chair, and Reviewers,

We sincerely thank the program committee and reviewers for their time, effort, and thoughtful feedback throughout the review process. We are also grateful for the reviewers’ unanimous recognition and positive recommendations of this work.

In this paper, we propose  ADAPT, a backpropagation-free test-time adaptation framework that models class-conditional feature distributions under a Gaussian assumption. This formulation enables efficient, one-pass adaptation via a closed-form solution, eliminating the need for iterative optimization and supporting both source-free online and transductive settings.

We are delighted that the reviewers appreciate and recognize the following strengths and contributions of this work:

* A simple yet compelling solution to test-time adaptation. **[uiWy, o7Us]**

* High efficiency in both time and memory by avoiding backpropagation and iterative optimization, enabling low-latency adaptation for real-time or resource-constrained scenarios. **[Hyj2, o7Us]**

* Broad applicability and flexibility, supporting both online and transductive settings without requiring source data. **[Hyj2, o7Us]**

* Extensive and convincing empirical evaluation across diverse benchmarks. **[uiWy, Hyj2, c5tn]**

* A well-motivated and well-written paper, presenting a principled method grounded in theoretical analysis. **[uiWy, c5tn]**


During the rebuttal and discussion phases, we were pleased to have *successfully addressed all concerns* raised by the reviewers. We will incorporate the valuable feedback into the final manuscript to further strengthen the paper. We sincerely thank all reviewers once again for their thoughtful follow-up and valuable guidance throughout the review process.


Sincerely,
The Authors

---

### Decision · Program_Chairs · 2025-09-17

**Decision:**

Accept (poster)

**Comment:**

ADAPT is a method for test-time adaptation, or updating on new data during testing to improve accuracy on distribution shifts, which does not require gradient optimization. Like other methods for TTA, ADAPT does not need the source data or offline access to the target test data. The experiments cover standard evaluations for test-time adaptation, like ImageNet variants including corruptions and renditions, and additional fine-grained image recognition benchmarks. The results rival or improve on existing methods with and without gradient optimization. Ablation and sensitivity analysis of the hyperparameters justify the inclusion of each part of the method.

Four expert reviewers with backgrounds on domain adaptation, transfer learning, meta-learning, and model editing vote for borderline acceptance (4, 4, 4) and borderline rejection (3). After the initial reviews the authors provided a rebuttal, and all reviewers then engaged in discussion and finalized their reviews. Following the rebuttal Hyj2 and uiWy raised their scores and o7Us and c5tn maintained their positive scores resulting in final ratings of one accept / 5 and three borderline accept / 4. While there is a shared concern about the weakness of the Gaussian assumption for class conditional distributions, there is nevertheless shared agreement on the thoroughness of the experiments and the positive results, and the choice of modeling is adequately addressed by the rebuttal. Further issues like missing related work (as highlighted by oyUs) and the possibility that results could be explained away by better prompting with a more recent model (as raised by c5tn) have also been addressed during the rebuttal. Once incorporated into the revision, the issues identified by review will have all been resolved. The area chair agrees with the final positive consensus of the reviewers on acceptance.